

# Challenges of constructing and selecting the "perfect" initial and boundary conditions for the LES model PALM

Jelena Radović[1,3], Michal Belda[1], Jaroslav Resler[2], Kryštof Eben[2], Martin Bureš[2,3], Jan Geletič[2], Pavel Krč[2], Hynek Řezníček[2], and Vladimír Fuka[1]

[1]Department of Atmospheric Physics, Faculty of Mathematics and Physics, Charles University, Prague, V Holešovičkách 2, 18000, Prague 8, Czech Republic
[2]Institute of Computer Science of the Czech Academy of Sciences, Prague, Czech Republic
[3]ATEM - Studio of ecological models, Prague, Czech Republic

**Correspondence:** Jelena Radović (radovic@karlov.mff.cuni.cz)

**Abstract.** We present the process and difficulties of acquiring the proper initial and boundary conditions (IBC) for the state-of-the-art LES based model PALM (Parallelized Large-Eddy Simulation Model). We use the mesoscale model WRF (Weather Research and Forecasting model) as a source of inputs for the PALM preprocessor, and investigate the influence of the mesoscale model on the performance of the PALM model. Sixteen different WRF configurations were used as a proxy for a multi-model ensemble. We developed a technique for selecting the suitable sets of IBC, performed PALM model simulations driven by them, and investigated the consequences of selecting a sub-optimal WRF configuration. The procedure was tested for four episodes during different seasons of the year 2019, evaluating WRF and PALM outputs against the atmospheric radio sounding observations. We show that the PALM model outputs are heavily dependent on the imposed IBC, and have different responses for different times of the day, and different seasons. We demonstrate that the main driver of errors is the mesoscale model, and that the PALM model is capable of attenuating, but not fully correcting them. The PALM model attenuates the impact of errors in IBC in wind speed, while for the air temperature, PALM shows variable behavior with respect to driving conditions. This study stresses the importance of high-quality driving IBC, and the complexity of the process of their construction and selection.

## 1 Introduction

Interest in studying the urban atmosphere and climate has been present since the last century and according to Mills (2014) it started with the work of Howard (1818). Due to the increasing number of city inhabitants and their impact on urban climate (Oke et al., 2017) and many other (scientific or commercial) relevant reasons (Souch and Grimmond, 2006), this field of study will remain the hotspot for researchers in the future. Characteristics of the urban areas and climate (e.g., Urban Heat Island (UHI), altered winds, air quality etc.) have been explored by many scientists (e.g., Arnfield, 2003; Mirzaei, 2015; Oke et al., 2017; Masson et al., 2020). Even though there are many challenges met while researching the urban areas (e.g., Arnfield, 2003; Blocken, 2018; Kubilay et al., 2020), there are several methods used for studying it (e.g., Blocken, 2015; Toparlar et al., 2017) among which are the Computational Fluid Dynamics (CFD) models.



A particular asset of the CFD method is that it allows detailed physics-based analysis of the urban climate and urban physical phenomena (Kubilay et al., 2020) i.e., for the scales below 2 km (Blocken, 2018). CFD models are versatile and
25 appropriate for e.g., studying flow around the buildings, pedestrian wind, vegetation cover related topics, etc. (e.g., Toparlar et al., 2017; Blocken, 2018). While using the CFD models for numerical simulations, one needs to consider which turbulence model to use. According to Blocken (2018) and Kubilay et al. (2020), CFD models mostly rely on the Reynolds Averaged Navier-Stokes (RANS), or Large Eddy Simulation (LES) turbulence models whose qualities and weaknesses have been the topic of many studies (see e.g., Hanjalic, 2005; Blocken, 2015, 2018; Maronga et al., 2019). In recent years, despite their
higher computational cost, LES models have become popular among researchers and modelers due to their higher accuracy and power to thoroughly capture the physical processes within the urban atmosphere.

Besides the large eddy simulation model for urban environments (uDALES; Suter et al., 2022), there is another LES-based state-of-the-art numerical model that allows scientists to study urban areas in a very high resolution called PALM (originally Parallelized Large-Eddy Simulation model) modeling system. Today's name PALM refers to the model name only, because it
was extended with a RANS core (Maronga et al., 2020). In general, the PALM model was the subject of many studies made for different purposes (e.g., Letzel et al., 2008; Resler et al., 2017; Heldens et al., 2020; Fröhlich and Matzarakis, 2020; Gehrke et al., 2021; Pfafferott et al., 2021). Despite the advantages and the level of detail it provides, there are still many limitations to it, some of which are mentioned in the earlier studies, but not in all its components and applications. One segment, which to the best of our knowledge has not been thoroughly investigated, is related to the issue of choosing the most suitable time-dependent
meteorological boundary conditions for running a given PALM model simulation.

While being capable of utilizing the standard cyclic boundary conditions (Maronga et al., 2015, 2020) and applying them to homogeneous and idealized setups (e.g., Gronemeier et al., 2017, 2021; Resler et al., 2017; Kurppa et al., 2018; Řezníček et al., 2023), PALM model offers the so-called one-way offline nesting system which enables it to take meteorological conditions from the mesoscale meteorological models and employ them throughout the entire PALM model simulation (Kadasch et al.,
2021). The application of such a system is most significant in the case of studying the atmosphere of a real, complex, and densely built urban environment. Furthermore, the utilization of as best and as realistic boundary conditions as possible during the PALM model simulations is of high importance in case of model validation studies and comparison against observations in which we strive to eliminate other possible sources of errors besides the model formulation and implementation (see e.g., Resler et al., 2021).

Up to now, several mesoscale model outputs have been used as drivers for the PALM model simulations, namely, Consortium for Small-scale Modeling (COSMO; Baldauf et al., 2011), Meteorological Cooperation on Operational Numerical Weather Prediction (MetCoOp) Ensemble Prediction System (MEPS; Bengtsson et al., 2017; Müller et al., 2017), ALARO/AROME (Termonia et al., 2018) and Weather Research and Forecasting (WRF) model (Skamarock et al., 2019), three of which (COSMO, MEPS, and ALARO/AROME) are not publicly available. Description of processing tools for initial and boundary conditions
(IBC) creation made for the COSMO model are described in Kadasch et al. (2021) and in Kurppa et al. (2020) for the MEPS modeling system. Furthermore, the ALARO/AROME has been used for PALM-4U initialization in a case study done by Zuvela-Aloise et al. (2022). On the other hand, given the fact that the WRF model is publicly available and widely used among





researchers, several different preprocessors have been developed: WRF_interface which is a part of the PALM distribution (Resler et al., 2021), and most recently by Vogel et al. (2022). Since this work covers the topic of finding the optimal choice

of WRF-modeled IBC, we only looked through the studies which employed WRF boundary conditions and their respective choices of WRF model setups (see e.g., Resler et al., 2017; Belda et al., 2021; Resler et al., 2021; Vogel et al., 2022). Each of these studies has implemented a particular WRF model setup which differs in the parameterization bundle, horizontal and vertical resolution, initialization data used (Global Forecasting System (GFS) or European Center for Medium-Range Weather Forecasts (ECMWF) atmospheric reanalysis of the global climate – ERA5; Hersbach et al., 2020) etc. When it comes to

the simulation periods considered in the available studies, a validation study by Resler et al. (2017) considered the heatwave episode which occurred during July 2015 as well as a sensitivity study by Belda et al. (2021), while Resler et al. (2021) selected several different episodes: two heatwave episodes during July 2018 and three episodes during November and December of the same year.

Studies like Belda et al. (2021) or Resler et al. (2021) are highlighting the need for having as accurate initialization data as possible for driving the PALM modeling system in validation studies. Radović et al. (2022) showed that the PALM model

results coincide with, and closely follow, WRF model outputs by comparing modeled vertical profiles against one another and testing their accuracy with the radio sounding data. Furthermore, by performing the standard statistical analysis, the same study stressed that different PALM model simulations show different quality of outputs for different variables (e.g., potential temperature, wind speed). Moreover, Vogel et al. (2022) say that the accuracy of the PALM-4U model output depends on the

WRF model setup.

Driven by these statements and bearing in mind that these studies utilized different WRF model configurations for driving their PALM model simulations, we designed and performed an experiment in which both different WRF configurations and different weather conditions were considered. As the main weather situations, we selected two events with high impact on urban environments, namely, a heatwave in July 2019, and a bad air quality episode in February 2019. To also include

non-extreme weather situations, we selected another two episodes during April and October with calm and stable weather. The WRF model was run in 16 different configurations producing an ensemble of 16 mesoscale simulations for each of the aforementioned weather situations. Our domain of interest is in the south-east part of the city of Prague, Czech Republic with its center in the vicinity of the Libuš meteorological station (LIBU; WMO ID 11520). This is a realistic urban area chosen to coincide with the aforementioned sounding station, enabling us to execute as realistic a comparison of the sounding profiles

as possible. We performed 14 PALM model simulations with identical model setups while only changing the IBC, i.e., the WRF ensemble member driving the PALM simulation. The main advantage of using a lower-resolution model as boundary conditions (compared to e.g. measured values) is the fact that they provide a physically consistent set of variables covering arbitrary locations. On the other hand, raw model outputs are inherently imperfect and an analysis can be used in their place, trading some physical consistency for a better agreement with observations. Individual bias correction for different variables

causes the same effect. Therefore we used raw WRF outputs in our study.

The first aim of this study is to show the complexity of choosing the optimal setup of IBC for the PALM simulations for a particular domain and a particular simulation period, and to show that many parameters must be taken into consideration





during the process. As a further matter, we try to recognize and separate the errors coming from the imposed IBC and the ones that originated from the microscale model.

This paper is structured as follows. Firstly, the choice of simulation periods is explained (Sect. 2.1). Secondly, the PALM model configuration is described in Sect. 2.2. The WRF model configuration, the ensemble members, and the selection strategy is presented in Sect. 2.3 and Sect. 2.4, respectively, followed by the result processing description (Sect. 2.5). The results are described in Sect. 3. Lastly, in Sect. 4, the discussion and future aspects are presented followed by the study limitations in Sect. 5, and conclusions in Sect. 6.

## 2   Methodology

### 2.1   Simulation periods

This experiment encompasses four different three-day episodes in February, July, April, and October of 2019. Furthermore, only the anticyclonic weather types are taken into consideration, since according to Zahradníček et al. (2022), these types are the most frequent ones occurring in the Czech Republic during the 1961–2020 period. Additionally, specific weather events we
take interest in i.e., bad air quality periods, heatwaves, stable and calm weather, coincide with and are a consequence of these weather systems. For instance, during winter (December, January, and February) they are characterized by clear and bright skies, with light wind speeds or no winds at all. These conditions, especially during the evening and nighttime, often lead to the occurrence of temperature inversion, a condition favorable for trapping the pollutants from vehicles, heating etc., creating bad air quality conditions within urban environments. Temperature inversion events are strongest during winter but can be observed
during the other three seasons as well. To give an example, the aforementioned phenomenon and bad air quality conditions within urban environments are used as an episode in a validation study by Resler et al. (2021). Similarly, in summer (June, July, August) when the sky is clear, the Sun is warming up the ground continuously, bringing hot and dry weather, which oftentimes, during anticyclonic conditions, leads to the creation of the weather event known as the heatwave (see e.g., Belda et al., 2021; Resler et al., 2021). Besides the intolerable temperatures that heatwaves bring to city inhabitants, the occurrence of increased
ground-level ozone concentrations in urban areas is another repercussion of this event (see e.g., Resler et al., 2021). Thereby, we take an interest in these two extreme weather phenomena for several reasons. Firstly, both have serious implications on city dwellers' health and well-being. Secondly, the heatwaves and bad air quality conditions are the two most important hazards for the city of Prague. And lastly, they are substantial for the PALM model's future validation (see e.g., Resler et al., 2017, 2021). Nonetheless, to broaden the study, and to see if the model behaves consistently throughout the year, we included two other
seasons (spring and autumn) without unique weather events. Hence, two more episodes have been selected, one in April and another in October. The choices of simulation periods in 2019 are as follows: 15–17 February (e1), 16–18 April (e2), 24–26 July (e3), and 20–22 October (e4). For detailed information about the meteorological conditions during selected episodes see Table 1.



**Table 1.** Meteorological conditions during the selected simulation episodes.

| episode | e1 | e2 | e3 | e4 |
|---|---|---|---|---|
| Weather type | Anticyclonic | Eastern anticyclonic | Anticyclonic, eastern anticyclonic | Southern anticyclonic |
| Minimum temperature | -2.4 °C | 1.3 °C | 15.9 °C | 7.6 °C |
| Maximum temperature | 14.0 °C | 19.0 °C | 38.5 °C | 16.5 °C |
| Cloudiness | Clear sky or passing clouds | Occasional sscattered clouds | Clear sky | Cloudy with fog |
| Precipitation | No | No | No | Sleet on 22.10.2019 during morning hours |
| Wind speed (up to 3000 m) | below 14 m s$^{-1}$ | below 14 m s$^{-1}$ | below 15 m s$^{-1}$ | below 18 m s$^{-1}$ |
| Wind speed (up to 10 m ) | below 3 m s$^{-1}$ | below 5 m s$^{-1}$ | below 3 m s$^{-1}$ | below 2.5 m s$^{-1}$ |

## 2.2 PALM model configuration

### 2.2.1 Model description and configuration

Simulations were done by the PALM modeling system (Maronga et al., 2020). The PALM model is based on the Large Eddy Simulation (LES) approach and it solves non-hydrostatic, filtered, Boussinesq-approximated, incompressible Navier-Stokes equations. The subgrid stress tensor is being modeled by the Deardorff (1980) 1.5-order closure involving Moeng and Wyngaard (1988) and Saiki et al. (2000) modifications. Pressure is calculated by a Poisson equation solved with the multi-grid scheme (description e.g. in Maronga et al., 2020). For spatial and temporal discretization, the upwind-biased 5th-order differencing scheme (Wicker and Skamarock, 2002), and the 3rd-order Runge–Kutta time-stepping scheme (Williamson, 1980) are employed, respectively. This core system is complemented by the so-called PALM for Urban Applications modules (PALM4U) specifically developed for studying the urban boundary layer and application to concrete problems i.e., city planning, urban climate studies etc. (Maronga et al., 2020). They include e.g. the land surface model (LSM; Gehrke et al., 2021), building surface model (BSM; Resler et al., 2017; Maronga et al., 2020), radiative transfer model and plant canopy model (RTM and PCM; Krč et al., 2021), human biometeorology module (BIO; Fröhlich and Matzarakis, 2020), online nesting (Hellsten et al., 2021) and the mesoscale nesting (MESO; Kadasch et al., 2021).

For the purposes of the experiment 14 simulations were conducted, and the length of each simulation episode was three days. Moreover, to accurately initialize the temperatures of the individual soil elements, building wall layers, and pavements, as well as the natural surface's soil moisture, our configuration included a one-day spin-up period. During the spin-up simulation, only the RTM, BSM, and LSM modules were used while the dynamic part of the model's code was switched off (see Maronga et al., 2020).



### 2.2.2 Input data and domain configuration

For solving the energy-balance equations as well as for radiation interactions, BSM, LSM, and RTM require the use of detailed
and precise input parameters describing the surface materials such as albedo, emissivity, roughness length, thermal conductivity, thermal capacity, and capacity and thermal conductivity of the skin layer. Urban and land surfaces as well as subsurface materials become very heterogeneous in a real urban environment when going to very fine spatial resolution. For this study three different data sources as input were used; (i) Copernicus Land Monitoring Service layer Urban Atlas 2018, (ii) OpenData platform of Prague Municipality (digital elevation model, building heights, etc.), and (iii) OpenStreet Maps as a source of
building locations outside of city of Prague. All datasets were processed to the static driver, an input file needed for the PALM model initialization (see PALM Input Data Standard - PIDS in PALM model documentation).

The domain used in this experiment is located in the southeastern part of Prague (see Fig. 1). In the central, northern, and north-western parts, the simulated domain is made up of diverse types of areas and includes all the typical objects which characterize an urban area (e.g., continuous and dense urban areas, transit roads, green urban areas, water bodies, etc.; Fig. 1).
The north-eastern part contains large green urban areas (code 1410 on Fig. 1). Moreover, the eastern and southern part is made up of arable land. The Vltava River crosses the domain in the south-north direction in the western part. Such land cover formation in the domain covers a diverse set of areas, chosen to challenge the model performance across the mentioned composition. Elevation in the domain varies between 171 and 381 m, mean elevation is 275 m. The highest hills are located in the southern parts of the domain. The Vltava River formed a deep valley in the western part, one small valley is located in
the center of the domain and a second larger one in the northern part, both of them forested. Slopes close to the valleys are steep and continuously changing to a plateau (see Appendix D). In the horizontal direction, the domain has a dimension of 8 x 8 km with 10 m horizontal resolution. Vertically, it extends up to the height of 2,830 m distributed on 162 vertical levels, 10 m resolution is applied until the 350 m height after which the stretching factor of 1.08 is implemented with the maximum stretching length of 20 m.

### 2.2.3 Initial and boundary conditions

The dynamic driver input file is used to supply the IBC for the PALM model. It consists of initial information for the entire domain and dynamic information about the time-dependent conditions at the boundaries. The 3D fields of potential temperature, velocity components (u, v, w), and water vapor mixing ratio originated e.g. from WRF model are horizontally and vertically interpolated to the PALM model grid. Since the PALM model has terrain represented in a higher resolution in comparison
to the WRF model, the vertical interpolation process incorporates stretching such that the bottom-level fields follow the fine terrain, while avoiding vertical distortion at the high levels as well as maintaining mass balance at the boundaries. Along with that, soil moisture and temperature, and a time series of large-scale surface forcing of surface pressure is taken from WRF and provided to PALM. Further guidance on data transformation and dynamic driver creation is available in Resler et al. (2021) and PIDS. The radiation variables i.e., downwelling shortwave (SW) and longwave (LW) are also taken from the WRF model
as they have a time resolution of 10 minutes. In addition, one physical phenomenon not resolved by the mesoscale model is



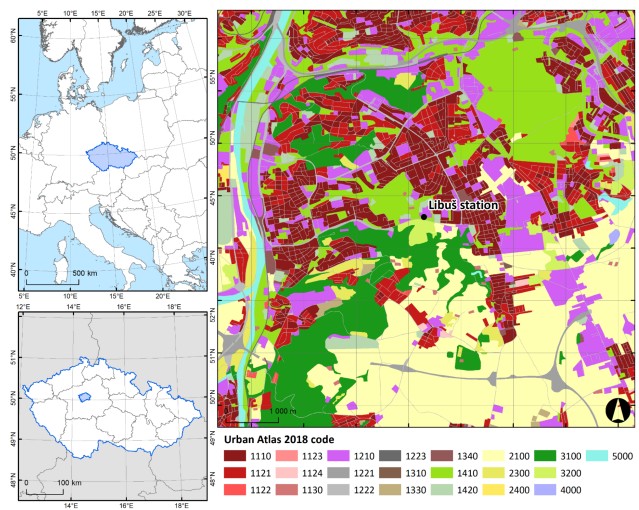

**Figure 1.** The location of the modeled domain in Europe (top left), and in the Czech Republic (bottom left). The map of the domain within the city of Prague with Libuš station (WMO ID 11520) location is presented at the right part of the figure. Land cover categories shown on the right are represented using Urban Atlas 2018 geodatabase with respective codes described in Urban Atlas (2018).

turbulence, thus it must be started artificially. This process is managed by the PALM synthetic turbulence generator (STG; Xie and Castro, 2008). Turbulence perturbations are forced into velocity components in the parent domain's lateral boundaries at every time step. All dynamic drivers used for this study were generated from the inner domain of the WRF model (Sect. 2.3).

## 2.3 WRF model configuration

The WRF mesoscale model (Skamarock et al., 2019) in version 4.4 was used to drive the PALM model through the PALM's mesoscale-nesting system. The model was run on two nested domains in horizontal resolutions of 9 km and 3 km with 49 vertical levels. For its initialization, ERA5 reanalysis was used. For the purposes of the experiment, an ensemble of 16 members was designed, with members differing in three factors (physics parameterizations) in which we expect an impact on the urban simulation. This design is "balanced" similarly to the statistical analysis of variance, i.e. every combination of factors is equally

represented. Thus we have:

- two versions of surface layer scheme (MM5 Similarity Scheme (Paulson, 1970), members 01–08 vs. Revised MM5 Scheme (Jiménez et al., 2012), members 09–16)

- two versions of PBL parameterization (Yonsei University (YSU) PBL scheme (Hong et al., 2006), members 01–04 and 09–13) vs. BouLac scheme (Bougeault and Lacarrère, 1989), members 05–08 and 13–16)

- four versions of urban parameterization (no urban parameterization, SLUCM - Single Layer Urban Canopy Model (Chen et al., 2011), BEP - Building Environment Parameterization (Martilli et al., 2002) and BEP+BEM - BEP in





combination with Building Energy Model (BEM; Salamanca and Martilli, 2010)). This factor rolls fastest, i.e. no urban parameterization in members 01, 05, 09, 13 etc.

Other parameterizations were in accordance with their common and widely used settings, e.g. NOAH LSM was used for all members. The Thompson scheme (Thompson et al., 2008) was used for microphysics for all ensemble members except the member 12 which required WRF single moment 5-class scheme (Hong et al., 2004) due to compatibility issues. The WRF ensemble simulation was performed for four episodes which amounts to 64 simulations all together. This design enables us to capture the eventual systematic effects of distinct parameterizations and it serves as a proxy for a multi-model ensemble of Numerical Weather Prediction models (NWP) thanks to its variability in model setup. For the summary of the experiment design see Table 2.

## 2.4 IBC selection workflow

Running a three-day simulation of PALM driven by each of the WRF ensemble members (i.e. 64 PALM model high-resolution runs) would be computationally expensive. Therefore a strategy of preselecting WRF ensemble members was developed to keep the computational costs low while satisfactorily sampling the variability. We classified the WRF simulations according to their performance in the representation of potential temperature and wind speed, evaluated against soundings taken at the Libuš meteorological station (WMO ID 11520) every day at 00:00, 06:00, and 12:00 UTC. The mentioned variables i.e., potential temperature and wind speed were chosen due to the fact that they are directly relevant for the PALM model's validation and are responsible for the development of atmospheric processes. The evaluation was based on root mean square error (RMSE) and correlation coefficient (r; Appendix A, cf. Resler et al. (2021) and Radović et al. (2022)). Two WRF ensemble members with the best and worst performance (closest to and farthest from the observations) were then preselected for the PALM runs. The performance, however, differs between variables (a similar issue was observed in Vogel et al. (2022) and Radović et al. (2022)), i.e. the best statistical values that some members showed for potential temperature were not the best for the same member in case of wind speed. Keeping in mind this behavior, the members with the lowest and highest RMSE values for temperature and another with the same characteristics but based on the wind speed are selected and the strategy was repeated for every one of the four selected periods. If two model members have similar RMSE values, the correlation coefficient may serve as a supporting statistical metric. The preselected configurations (coded by member numbers) are summarized in Table 3. Some configurations have multiple occurrences.

To support this method of selection, a series of descriptive statistics were computed to assess the effects of factors represented by PBL, surface layer and urban physics. No systematic superiority of one parameterization over another was detected. The effects which were observed were the following:

- for the October episode, the Boulac PBL outperforms the Yonsei PBL

- for the February episode, the SLUCM is systematically the worst urban parameterization

Since no single effect was capturing the differences in performance, we preferred the selection method described above.





**Table 2.** Summary of the experiment design - parameterizations in WRF model ensemble members.

| WRF member ID | 01 | 02 | 03 | 04 | 05 | 06 | 07 | 08 |
|---|---|---|---|---|---|---|---|---|
| Surface layer | MM5 Sim. Sc. | MM5 Sim. Sc. | MM5 Sim. Sc. | MM5 Sim. Sc. | MM5 Sim. Sc. | MM5 Sim. Sc. | MM5 Sim. Sc. | MM5 Sim. Sc. |
| PBL | YSU | YSU | YSU | YSU | BouLac | BouLac | BouLac | BouLac |
| Urban Physics | 0 | UCM | BEP | BEM | 0 | UCM | BEP | BEM |
| WRF member ID | 09 | 10 | 11 | 12 | 13 | 14 | 15 | 16 |
| Surface layer | Revised MM5 | Revised MM5 | Revised MM5 | Revised MM5 | Revised MM5 | Revised MM5 | Revised MM5 | Revised MM5 |
| PBL | YSU | YSU | YSU | YSU | BouLac | BouLac | BouLac | BouLac |
| Urban Physics | 0 | UCM | BEP | BEM | 0 | UCM | BEP | BEM |





**Table 3.** The WRF ensemble member ID numbers selected for each simulation episode.

| Episode | Potential temperature | | Wind speed | |
|---------|---------------------|--------------|--------------|--------------|
|         | Lowest RMSE | Highest RMSE | Lowest RMSE | Highest RMSE |
| e1 | 03 | 02 | 16 | 10 |
| e2 | 05 | 14 | 09 | 14 |
| e3 | 01 | 12 | 07 | 12 |
| e4 | 05 | 14 | 07 | 01 |

## 2.5 PALM model near-surface output processing

The PALM model near-surface outputs methodology processing performed for air temperature at 2 m, wind speed at 10 m, Mean Radiant Temperature (MRT), Physiological Equivalent Temperature (PET), and Universal Thermal Climate Index (UTCI) is done as follows. At first, the selection of the averaging periods is made to distinguish between parts of the day influenced by the solar input. Hence, four different times of the day are considered, i.e., morning (1 hour before sunrise), solar noon (30 minutes before to 30 minutes after solar noon, referred to as noon in the figures), daytime (between sunrise and sunset), and the

nighttime (between sunset and sunrise). The selected hours are adjusted according to the season for which the simulation was performed and are displayed in the Universal Time Coordinated (UTC) time standard. Regarding the simulation results, only the differences between the PALM members driven by the WRF members with the lowest (the best) and highest (the worst) RMSE values are shown. The basic statistics obtained for both WRF and PALM model outputs are shown in Appendix C. Each table demonstrates spatial minimum, average and maximum of three-day averages for each variable, period and selected

member. It must be noted that the PALM model 2 m air potential temperature values used for calculation and shown in the following figures is estimated from the logarithmic interpolation due to the fact that the first prognostic grid point is placed at the height of 5 m (see PALM model documentation). The same principle for a horizontal component of the wind speed at 10 m is used, while the vertical component is calculated accurately for each grid point in the modeled domain. The WRF model air temperature and wind speed outputs used for the near-surface comparison are taken from the lowest level available in the

model. The lowest level WRF model outputs are used since they are utilized for the PALM model IBC creation.

## 3 Results

### 3.1 Vertical structure

In this section, we compare the potential temperature and wind speed vertical structure for both WRF and PALM models with radio soundings. WRF vertical profiles are taken from the grid box closest to the Libuš station, while PALM vertical profiles

are averaged over a 10 x 10 grid box area around the center of the domain. The PALM profiles are averaged due to the fact that





the WRF grid cell is significantly larger than the PALM grid cell. Comparison of the vertical profiles is performed at the times of radio soundings collection (00:00, 06:00, and 12:00 UTC).

Before diving into a detailed analysis of individual episodes, we present an overall view of the simulations in terms of errors with respect to the soundings. The results for all PALM simulations for profiles for the first 300 m of the atmosphere are 250 summarized in Fig. 2. Every point representing a simulation is marked as "Best" or "Worst" according to the WRF ensemble member selection in Table 3. Also, the criteria used for selection are distinguished by color, and all the Best/Worst pairs are connected. Thus we can identify the improvement/deterioration of the RMSE when going from WRF to PALM (distance to the dashed diagonal line), as well as the improvement/deterioration when changing the parameterization of WRF (the connected point). Since the majority of the points lie under the diagonal, we can argue that the detailed modeling of PALM mostly brings 255 an improvement in the vertical profile covering the first 300 m of the atmosphere. The positive effect is more evident in cases where the error in the WRF simulation is large. It is also seen that the effect of selecting a less appropriate parameterization in WRF can have a large impact on the error. Nevertheless, a corrective behavior of the PALM simulation is evident even in these cases. On the other hand, if the error in the WRF simulation is relatively small, we can not claim a systematic improvement in the vertical profile, brought by the PALM simulation.

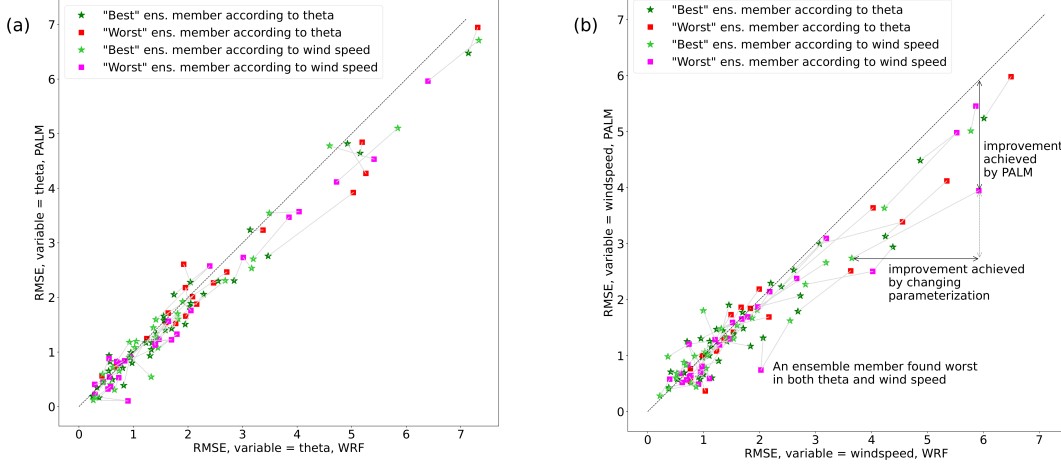

**Figure 2.** Scatter plots of PALM and WRF simulation RMSE values for potential temperature (a) and wind speed (b) vertical profiles for the first 300 m of the atmosphere and all preselected WRF model ensemble members, as well as for all PALM model simulations performed.

Next, we analyze the simulations in more detail for the two distinct seasons. The PALM model statistical metrics are calculated with respect to radio soundings for the atmospheric layer of up to 3000 m a.s.l. (Appendix B). In the winter episode, during night and morning, potential temperature vertical profiles from the PALM simulations do not deviate from the WRF modeled profiles but are slightly warmer or colder than WRF profiles in the lower layers (Fig. S02a, c–d, f–g, and i). During





the midday sounding hour, the PALM profiles follow the WRF profiles closely in both lower and higher atmospheric layers

(Fig. S01–S02beh). In some cases, PALM simulations show added value over their driving conditions from WRF, thus being closer to the observations (e.g., Fig. S02c–d). Overall, the shape of the WRF potential temperature profile is captured by the PALM model. The atmospheric stability/instability is represented well by PALM. The PALM wind speed vertical profiles, in general, follow the WRF modeled vertical profiles (Fig. S03–S04), but in some cases can deviate from them in the layers near the surface during morning and midnight hours (e.g., Fig. S04c–d and i). On the other hand, during the 12:00 UTC times,

the agreement between PALM and WRF is much larger in the layers close to the surface. Altogether, compared to the potential temperature, wind speed vertical profiles show larger discrepancies between WRF and PALM. Differences between the best (03), and the worst (02) WRF ensemble members' potential temperature vertical profiles are pronounced the most during 12:00 UTC sounding hours (Fig. S02beh). On the other hand, the differences between the best (16) and the worst (10) WRF members' wind speed vertical profiles are more pronounced across all sounding times (Fig. S04). The statistical analysis for

the profile up to 3000 m a.s.l. shows that the PALM simulations with respect to the WRF model potential temperature have lower RMSE in case of the member 03, and higher for the rest of the members. In the case of the wind speed, the PALM model RMSE values are higher than all of their corresponding WRF ensemble members (Table A1 and B1).

In the summer episode, potential temperature vertical profiles from the PALM are consistent with the respective WRF profiles and show the highest consistency during 12:00 UTC (see Fig. S09–S10beh). As in the e1 episode, the PALM shows

the added value over the WRF driving conditions which is seen for the member 12 (Fig. S10h–i). The shape of the PALM profiles follows the WRF profiles, but smaller discrepancies can be seen in the lower atmospheric layers. The atmospheric stability/instability is well captured by the PALM during all sounding times except for one sounding time (see Fig. S10g). Similarly to the wind speed profiles in e1, in this episode, the PALM profiles generally stay consistent with the WRF profiles, but the largest discrepancies can be seen during nighttime sounding hours (Fig. S12cfi). The differences between the best (01)

and worst (12) WRF simulations are not pronounced in the potential temperature profiles (Fig. S06). In the case of the WRF wind speed profiles, the differences between the best (07) and the worst (12) members are more noticeable (Fig. S08). The RMSE values calculated for the PALM potential temperature profiles for the e3 episode are lower in comparison to their WRF member pairs. On the other hand, for the same episode, the RMSE calculated for the wind speed profiles is higher for the PALM members (Table A3 and B3).

In summary, the highest consistency between the PALM models' vertical profiles and the corresponding WRF model profiles is seen during 12:00 UTC, and this behavior is valid for all simulation episodes. In general, the shape of the WRF profile is followed by the PALM profile, and most of the differences between the PALM and WRF vertical profiles are seen in the layers near the surface. In certain cases, the PALM introduces an added value to driving WRF conditions (see Fig. S10b and h–i). The wind speed vertical profile comparison is more chaotic and depending on the simulation period and the sounding time

the correspondence between the PALM's and the WRF's profile can vary. In general, the differences between them are highest in the layers closest to the surface, which is to be expected since the terrain representation is different in these two models. This behavior has already been seen in the work done by Resler et al. (2021). The statistical analysis performed on the PALM vertical profiles showed that in the case of e1, e3, and e4, RMSE values obtained for the wind speed are higher for PALM than





for WRF, while for the e2, they are lower. The PALM RMSE values for the potential temperature vertical profiles are lower
than the WRF RMSE values in the case of e2, e3, and e4 episodes, and that for the three (02, 10, 16) members is higher for
the PALM in episode e1. This analysis has a limiting factor which is related to having radio sounding observations only three
times per day, thus preventing us from performing more robust statistical and qualitative analysis.

### 3.2 Near-surface evaluation

The second set of results refers to the influence of the selected pairs of the IBC on temperature at 2 m and wind speed at
305 10 m. For the July episode, MRT, PET, and UTCI indexes were also computed. The figures depict the best/worst difference
for the two WRF members and the difference of corresponding PALM members driven by them. For each daytime period we
display the WRF field and the corresponding PALM field. Note that the PALM domain is represented by a red square in the
WRF field and the figure thus illustrates the downscaling of a couple of WRF gridpoints to a much higher resolution and, in
particular, the amplification or attenuation of IBC differences by the downscaling process executed by PALM. In the text, the
310 results for February and July are presented. Results for the other two episodes are deferred to the Supplement. The best/worst
classification is based on potential temperature.

In Fig. 3, differences in the air temperature between members 03 and 02 for the February episode, and four different aver-
aging periods (morning, noon, daytime, nighttime) are presented. For all periods we can see a qualitative consistency between
the WRF gridpoint values and PALM fields. The downscaling process done by PALM exhibits a distinct suppressive behavior
which can be attributed to the fact that both sets of IBC undergo the same local processes. On the other hand, the added value
brought by the LES model, in particular by the high-resolution topography, surface representation, and resolved turbulence, is
clearly seen. The local processes naturally enlarge the differences within the fields and these differences are often "transported"
to different locations compared to the WRF field. Also, note the fringe-like pattern in Fig. 3a, with both positive (0.5 K to 2 K)
and negative (-0.5 K to -2 K) differences appearing across the domain. The same effect is seen in Fig. 3f, although it is less
pronounced. This may be attributed to different urban parameterizations in the WRF members. In addition, rough transitions in
the driving fields may promote the generation of waves in the microscale model. The argumentation above is supported by the
descriptive statistics in Table 4, where we can see lower average differences in the PALM fields but mostly higher minimum
and maximum differences.

Differences in the air temperature for four different averaging periods (morning, noon, daytime, and nighttime) during the
325 July simulation episode between the members 01 and 12 for WRF model and PALM model are shown in Fig. 4. The overall
behavior of the downscaling process shows similar effects as in the February episode. This is also confirmed by the values in
Table 5. The influence of the orography and landuse on the differences is seen here more clearly than in the February episode.
One noteworthy feature is that in the summer episode, differences between the two WRF simulations are on average more
pronounced during the night, while in February, nighttime differences are higher. Exploring this behavior is beyond the scope
of this manuscript, however, in terms of the influence on the high-resolution simulation, PALM follows this behavior.

For both February and July episodes, it is clear that the differences in the WRF fields are the largest in the urban area. Since
all four members share the YSU parametrization of PBL, the differences have to be attributed to urban parameterization, which



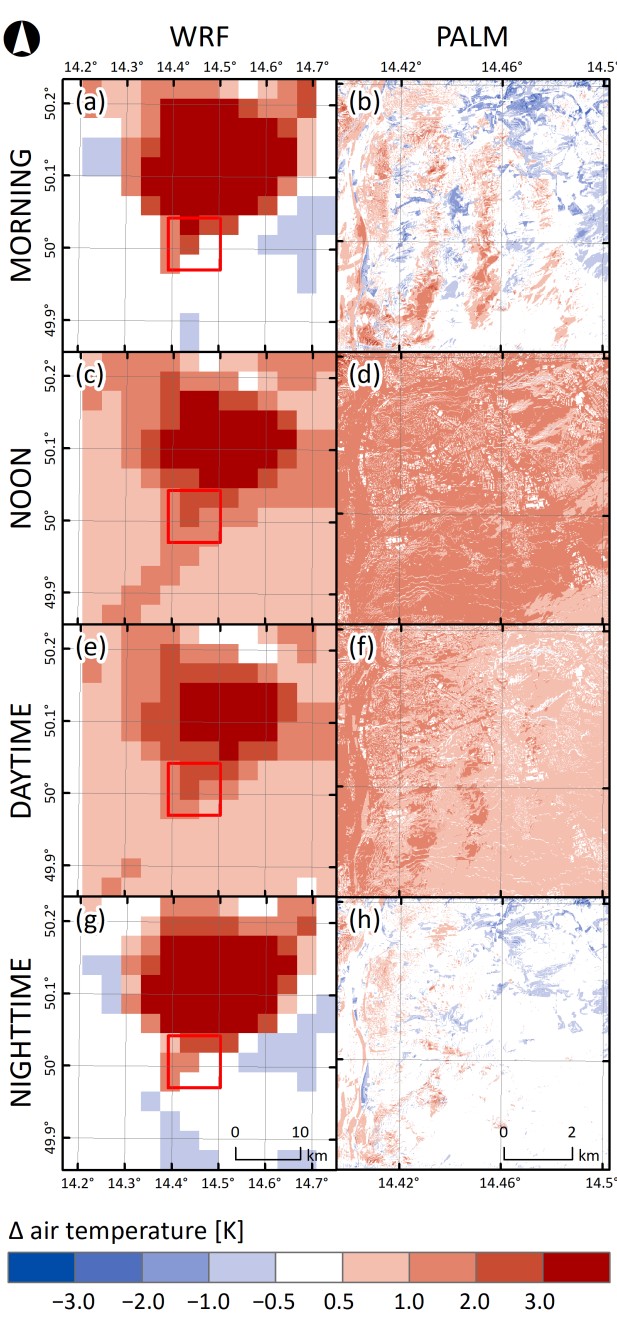

**Figure 3.** Differences between three-day averages of air temperature for four selected time periods (morning, noon, daytime, and nighttime) taken from the WRF and PALM model members 03 and 02 for the February episode. The first column refers to the difference between the best (03), and the worst (02) WRF model members selected based on the potential temperature, the second column refers to the difference between the PALM model members driven by the said WRF model members. The PALM model simulation domain is depicted with the red square.





**Table 4.** February episode minimum (min), average (avg), and maximum (max) differences in air temperature for four different averaging periods (morning, noon, daytime, and nighttime) between the members 03 and 02 for WRF and PALM model. In PALM fields the differences of the air temperature at 2 m are taken from the 2D 10-minute averaged files, while for WRF fields the air temperature from the lowest model level was taken.

| Air temperature [K] | WRF | | | PALM | | |
|---|---|---|---|---|---|---|
| | min | avg | max | min | avg | max |
| Morning | -0.18 | 1.77 | 3.00 | -3.97 | 0.08 | 3.29 |
| Noon | 1.29 | 2.14 | 2.67 | -1.32 | 1.13 | 3.03 |
| Daytime | 1.02 | 1.89 | 2.41 | -1.93 | 0.87 | 2.01 |
| Nighttime | -0.42 | 1.55 | 2.88 | -4.24 | 0.04 | 2.80 |

is 0 vs. BEM in the July episode and BEP vs. UCM in the February episode (cf. Table 2). These differences in the WRF fields thus propagate into the microscale simulation.

**Table 5.** July episode minimum (min), average (avg), and maximum (max) differences in air temperature for four different averaging periods (morning, noon, daytime, and nighttime) between the members 01 and 12 for WRF and PALM model.

| Air temperature [K] | WRF | | | PALM | | |
|---|---|---|---|---|---|---|
| | min | avg | max | min | avg | max |
| Morning | -1.69 | -1.37 | -0.99 | -2.71 | -0.59 | 1.98 |
| Noon | -1.52 | -1.15 | -0.57 | -4.23 | -0.48 | 2.15 |
| Daytime | -1.01 | -0.74 | -0.40 | -1.26 | -0.45 | 2.99 |
| Nighttime | -3.76 | -2.46 | -0.56 | -3.58 | -0.99 | 2.03 |



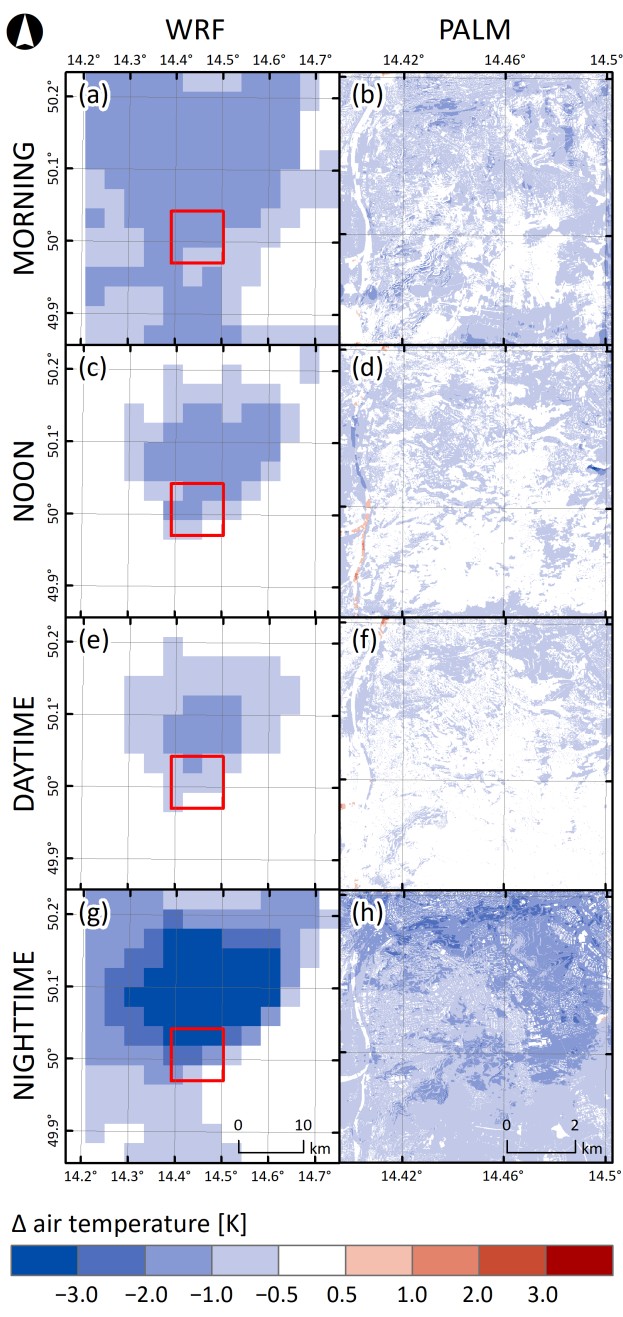

**Figure 4.** Differences between three-day averages of air temperature for four selected time periods (morning, noon, daytime, and nighttime) taken from the WRF and PALM model members 01 and 12 for the July episode. The first column refers to the difference between the best (01), and the worst (12) WRF model members selected based on the potential temperature, the second column refers to the difference between the PALM model members driven by the said WRF model members. The PALM model simulation domain is depicted with the red square.





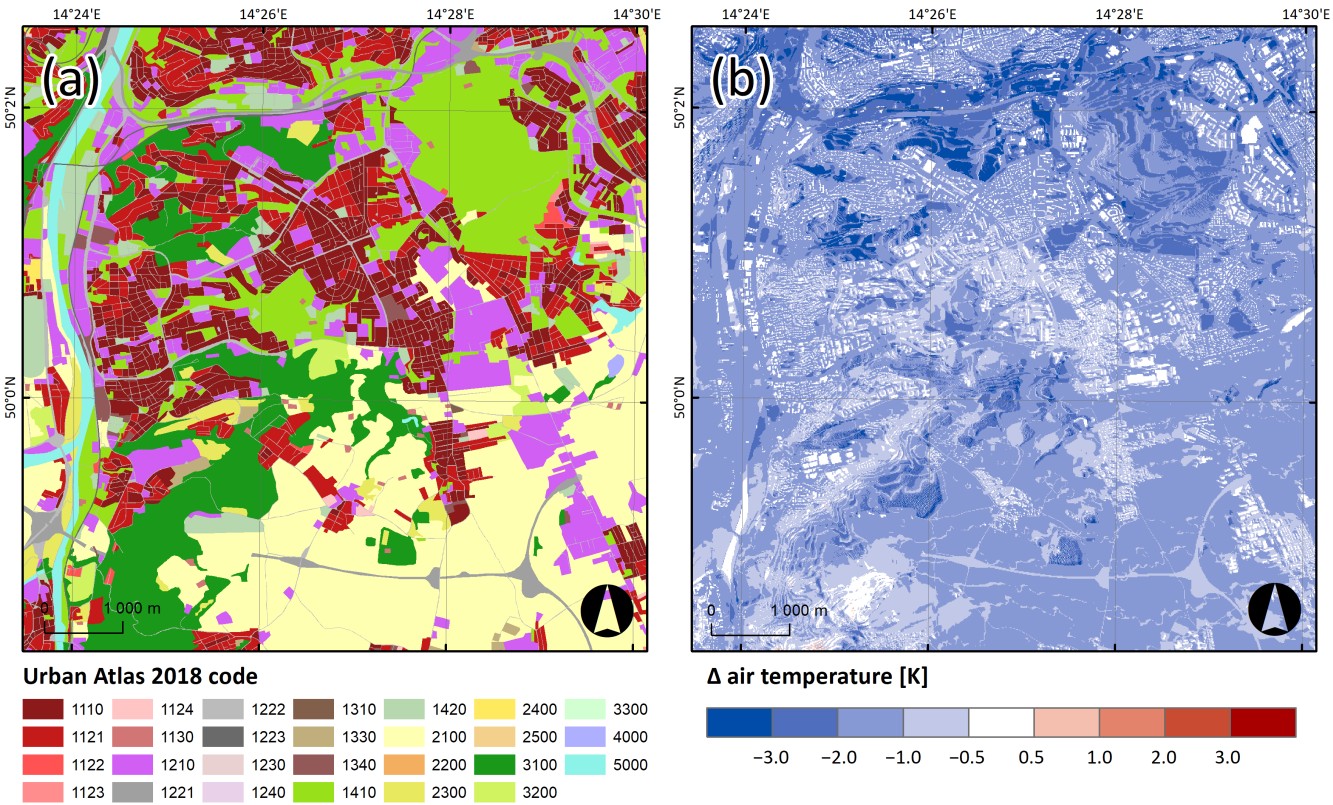

**Figure 5.** Differences between three-day averages of air temperature for July nighttime period simulated by PALM model members 07 and 12 (b), and landuse (a).

For convenience, we provide here a detailed look at one of the maps together with the landuse in Fig. 5. The PALM output on the Fig. 5b for the nighttime averaging period is taken from the pair 07-12 chosen based on the statistical analysis for the wind speed. This comparison shows that the differences between the two simulations at the local scale are driven mainly by the difference in landuse and thus the full difference is a composite of large-scale and local-scale forcings.

From the results of the two episodes, we can conclude that the PALM differences are altogether analogous to the WRF differences as seen for example in the case of e3 episode for all averaging periods (Fig. 4). The same conclusion applies for the e1 episode in the case of noon, and daytime averaging periods (Fig. 3c–f). However, for the morning, and nighttime period for the e1 episode, PALM introduces negative differences which range from 0 K to -1 K which are especially visible during the morning averaging time. For both presented episodes, PALM differences are on average lower and attenuated in comparison to the corresponding WRF differences. Moreover, the average differences for all averaging times are lower for PALM, especially during the morning and the nighttime averaging period for the e1 episode where they have values of 0.08 K and 0.04 K, respectively, which means that PALM, in general, does not amplify the differences across the domain (see Table 4–5). With respect to the wind speed, the attenuation of the differences is more pronounced, especially during the daytime and noon




averaging period across all simulation episodes (see e.g., Fig. S19, S23, S27, S31). The PALM differences are consistent with the given driving field, i.e., if one WRF ensemble member is warmer or colder, the same member will be warmer or colder in

the PALM model as well. On average, PALM tends to diminish the differences (Fig. 4f and Fig. S21c–g), but it does take over, and amplifies them on certain surfaces such as water bodies (see Fig. 1, surface code 5000), where a certain nonlinearity in the response exists, and PALM creates its own structures (see also e.g., Fig. 4d, and Fig. S25cdf).

To provide a summary view on the PALM model's sensitivity to the IBC, an additional analysis was performed in which

the spatially averaged 1h average differences of air temperature and wind speed are analyzed between all the PALM outputs. The analysis is presented for the e1 and e3 episodes in Fig. 6–7, respectively while the rest of the analysis is included in the Supplement as Fig. S35–S36.

As for the time course of the PALM differences themselves ((b), (d), (f), (h) in Fig. 3–4), a time pattern may occur. In case of the e1 episode (Fig. 6), the average differences show a diurnal cycle. This pattern is more prominent in the case of air

temperature (Fig. 6a) where differences start to increase around 06:00 UTC until approximately 14:00 UTC. On the other hand, the average differences calculated for the wind speed are low most of the time and start increasing only at the end of the second day of the simulation. This diurnal pattern is present for the e3 episode as well with slightly larger magnitudes of differences than in the e1 episode (Fig. 7). The average differences have a shorter period of increase lasting from 17:00–00:00 UTC for both air temperature and wind speed (Fig. 7)

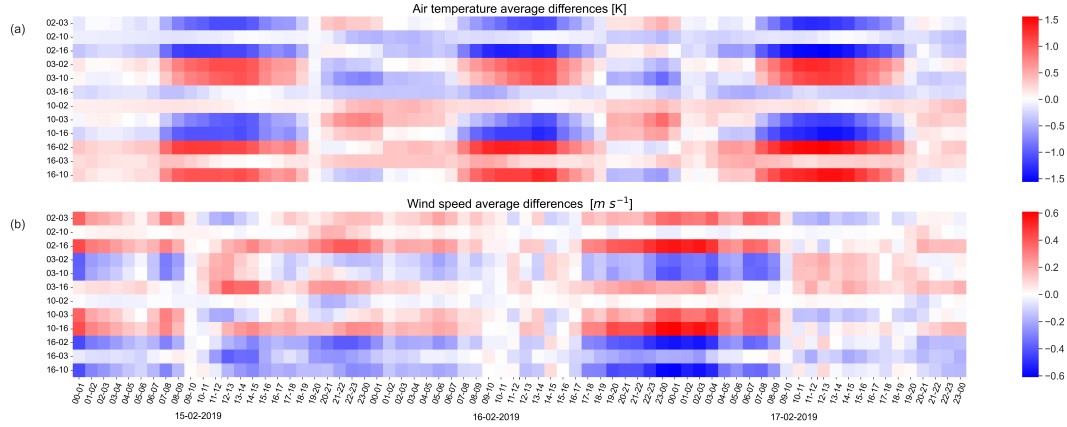

**Figure 6.** Spatially averaged 1hr average differences of air temperature (a) and wind speed (b) calculated for all the combinations taken from the PALM model outputs for the e1 episode. On the x-axis, the averaging hours in UTC along with the simulation dates are presented, and on the y-axis are the ID numbers of PALM model differences.

.





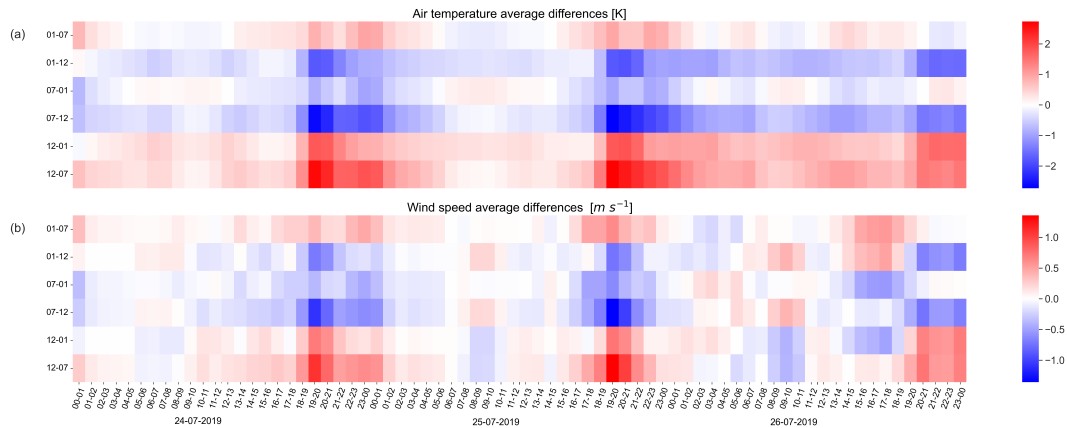

**Figure 7.** Spatially averaged 1hr average differences of air temperature (a) and wind speed (b) calculated for all the combinations taken from the PALM model outputs for the e3 episode. On the x-axis, the averaging hours in UTC along with the simulation dates are presented, and on the y-axis are the ID numbers of PALM model differences.

It is apparent from Fig. 6–7 that during certain hours differences between two PALM simulations driven by a different set of IBC is relatively small. This further means that the effect of the IBC is not large during this time and that the local processes resolved by the high-resolution PALM model are able to suppress their influence.

### 3.3 Influence of the IBC on the biometeorological indexes during e3 episode

The PALM model has been proven to be of service for urban planning and the UHI mitigation strategies (e.g., Belda et al., 2021), and it is for example applied in the works of Geletič et al. (2022, 2023) for heat stress mitigation strategies. One of the major conclusions of Belda et al. (2021) was that PALM can show opposite sensitivity between physical and biophysical temperature indicators. The differences in the UTCI index obtained from the PALM model members for the e3 episode (Fig. 8) show a consistent response in temperature and UTCI. The strongest minimum average difference has the pair 01-12 for the

nighttime averaging period (-4.6 K), the strongest differences are present near the north and west boundary of the simulation domain. The highest maximum average difference for the noon averaging period (1.5 K). The average difference for all 4 periods is around -0.5 K. The pair 07-12 shows similar behavior for the nighttime (-4.1 K), but with more significant effects of elevation differences visible close to deep valleys predominantly (see Appendix D). The highest maximum average difference was found for the morning averaging period (1.8 K). The average differences obtained for UTCI do not differ much from the

averaged values obtained for the air temperature (Table 5 and C5); this behavior is valid for both pairs (see Table 6).



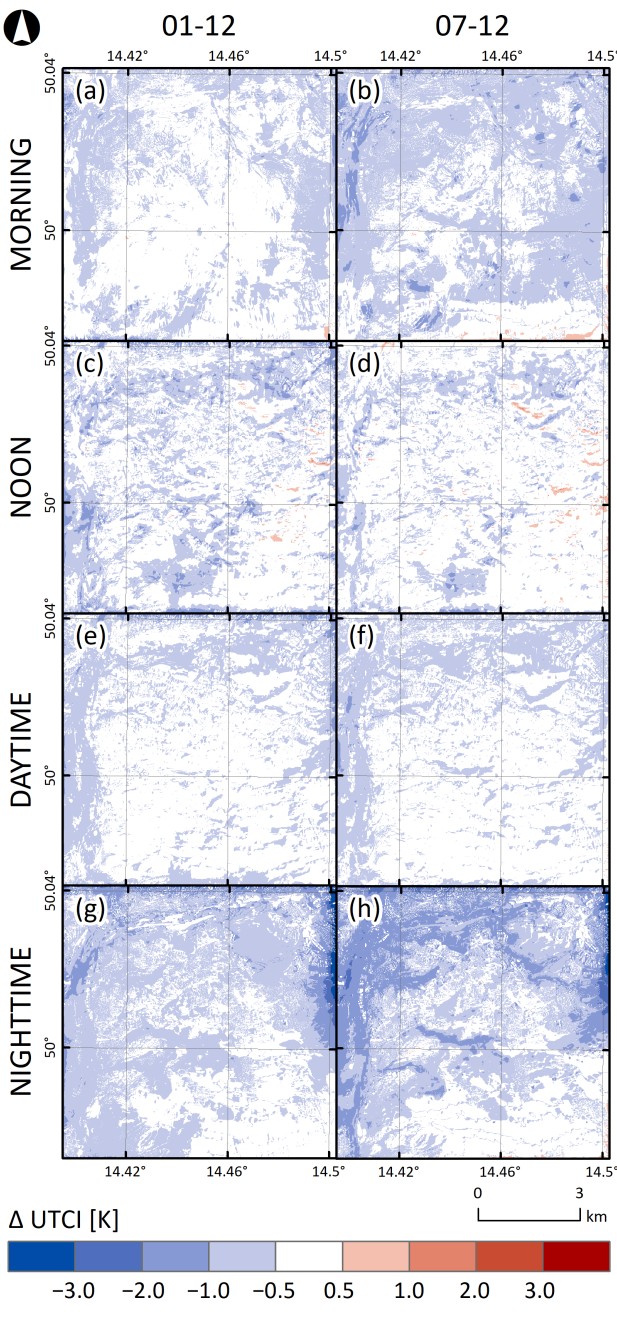

**Figure 8.** Differences between three-day averages of UTCI for four selected time periods (morning, noon, daytime, and nighttime) taken from the PALM model simulations for the July episode. The first column refers to the difference between the best and the worst member selected based on the potential temperature (01-12), the second column refers to the difference between the best and the worst member selected based on the wind speed (07-12).



**Table 6.** July episode minimum (min), average (avg), and maximum (max) differences in UTCI for four different averaging periods (morning, noon, daytime, and nighttime) between members 01 and 12, and members 07 and 12 for the PALM model.

| UTCI [K] | PALM 01-12 | | | PALM 07-12 | | |
|---|---|---|---|---|---|---|
| | min | avg | max | min | avg | max |
| Morning | -2.60 | -0.45 | 1.36 | -2.48 | -0.58 | 1.76 |
| Noon | -2.76 | -0.46 | 1.54 | -1.91 | -0.31 | 1.69 |
| Daytime | -2.20 | -0.50 | 0.37 | -1.54 | -0.46 | 0.85 |
| Nighttime | -4.58 | -0.68 | 0.65 | -4.08 | -0.78 | 1.51 |

## 4 Discussion and future aspects

Resler et al. (2021) showed that in order for a PALM simulation to be realistic, good quality of input data (e.g., static driver data, mesoscale forcing, etc) is necessary. That study also indicates that the errors occurring in the mesoscale model propagate into the PALM simulation. Thus a question is raised, namely, to which extent the driving conditions could be the main cause of potential errors and inconsistencies in the PALM model outputs.

Our validation of the vertical profiles confirms the importance of the driving conditions. For potential temperature PALM profiles have, in general, lower RMSE than the driving WRF ensemble members (episodes e1-member 03, e2, e3) or similar RMSE (episode e1- WRF member 02, e4), see Appendix B. On the other hand, the PALM RMSE values obtained for wind speed are higher (episode e1, e2- WRF member 09, e3, e4) or similar (e2 member 14).

Among the members of the ensemble of sixteen different WRF model realizations differing in urban parameterization, PBL parameterization and surface layer parameterization (Table 2), no specific setting can be marked as uniformly better than the rest, not even for any specific season. From a theoretical standpoint, all combinations are acceptable. If we base the comparison on the statistical metrics for one variable (e.g., potential temperature or wind speed), the results are not consistent across seasons. For a specific season, the selection of best/worst ensemble member gives different results when based on potential temperature or wind speed. Some degree of inferiority is seen in the member 14 (BouLac PBL, UCM urban par.) though, which in three out of the eight cases has the worst RMSE value, namely, for seasons e2 and e4 in potential temperature, and in season e2 in wind speed (Table 3). In e2 it has the highest RMSE for both variables. Therefore, in order to determine whether the WRF model or a specific WRF model realization performs better or worse for a certain season, a certain variable, and if it shows any kind of long-term consistency in general, an exhaustive long-term analysis has to be performed in advance.

A natural consequence of the impact of the IBC on the microscale simulation is the fact that any validation involves the model couple mesoscale/microscale. By model couple, we mean the driving mesoscale model (e.g., WFR, ICON or COSMO, and transitively their driving data), and the high-resolution model i.e., PALM. Thus, while providing the driving data to the microscale model, the errors and uncertainties coming from the mesoscale model are introduced and their magnitude or origin





is not known. So, without separating the errors which arise from the mesoscale and microscale models, one could be deceived,
and could possibly find some microscale model processes erroneous or wrongly represented, while the true origin of the errors
comes from the mesoscale model and driving fields it provides. Hence, further development of the microscale model (e.g.,
PALM) could be targeting the wrong part or a process, and consequently, introduce overcorrecting model adjustments, but in
the end, getting better results for the wrong reason. Furthermore, Belda et al. (2021) tested the sensitivity of the PALM model
to potential erroneous material parameters settings in which they showed that PALM model temperature shows the highest
sensitivity of $\pm 0.18$ K to the setup of certain building and material parameters (e.g, albedo, emissivity). Compared to the
mentioned study, the variability in response to near-surface temperature introduced by different driving conditions shown in
this study is much higher than the variability coming from the surface parameters ($\pm 3$ K), thus proving the PALM model's high
sensitivity to IBC. This observation needs to be considered especially in the process of model validation. The influence of the
imperfect boundary conditions on the results might lead to the tuning of the local model by changing the internal parameters
to achieve better correspondence with observations, in effect "getting good results for the wrong reasons".

The WRF model used in this study is not able to explicitly resolve the large eddies which have a strong impact on the
atmospheric flows, momentum, heat, and air pollution transport in the boundary layer, while on the other hand, the PALM
model can. But, despite the assets of the PALM model, its results are largely dependent on the quality of the mesoscale
WRF simulation. This is a principle usually known as "garbage in, garbage out" in many fields, such as limited area regional
modeling, in which the regional models cannot correct large-scale errors imposed from the lower-resolution driving models
(e.g. Giorgi, 2019). An integral future perspective of this work is related to the coupling of the PALM model with more
mesoscale models, namely, Icosahedral Nonhydrostatic Weather and Climate Model (ICON; Zängl et al., 2015), ALADIN
model etc., and validating the mesoscale-microscale model couple against the observational data.

## 5   Study limitations

The work presented here, PALM model configuration, and input data used, have certain limitations which are listed in the
following paragraphs.

- The PALM model simulations are conducted only for specific three-day periods. The main reason is that PALM simula-
  tions are computationally expensive. These three-day periods, even though conducted for four episodes throughout the
  year, might not be sufficient to assess the full influence of the WRF model boundary conditions on the PALM model
  response and its results.

- The resolution used for the simulations is 10 m, and no nested domain in higher resolution (e.g., 2 m) is utilized. Such
  choice of resolution can potentially mask certain phenomena, and thus influence the assessment of the influence of the
  initial and boundary conditions to the PALM outputs. Moreover, one cannot see how the higher resolution domain would
  behave with respect to the driving conditions, nor if it would modify the driving fields in any aspect.



- Due to the technical error during the process of static driver generation there is a mismatch between the realistic terrain height and the terrain height used in the simulations. Thus the simulation terrain height is shifted down by 10 m with respect to sea level. To be sure that this shift does not affect the results, the e1 episode simulations were repeated. No substantial differences were observed, qualitatively or quantitatively.

- For the purpose of this study we only used a particular sample of the WRF model outputs, thus not utilizing the full ensemble of the produced outputs due to the extremely high computational costs of the LES-based PALM model simulations.

- The WRF model in version 4.4 utilized for this study uses Moderate Resolution Imaging Spectroradiometer (MODIS) dataset. However, Demuzere et al. (2023) recently developed, and implemented a hybrid 100-m global land cover data set for the WRF model. Such advancement in the resolution of a land cover can be important for urban modeling applications, and consequently change the behavior of the initial and boundary conditions produced by the WRF mesoscale model, further influencing the PALM model outputs.

- This study is a case study performed for the city of Prague, Czech Republic. In order to confirm the behavior and influence of the boundary conditions on the PALM model simulations more case studies are necessary. Regardless, these results are applicable to the PALM model performance with regard to the IBC in general.

- Due to the prevailing anticyclonic weather type typical for the city of Prague and the Czech Republic in general, this study is limited to the aforementioned weather type. In addition, testing of the WRF model performance with respect to weather types has been performed (not shown), and the conclusion obtained from this analysis shows that the WRF model performance has no systematic behavior with regard to weather types, meaning it does not perform better or worse for e.g., cyclonic or anticyclonic weather systems.

- Another limitation which has to be mentioned, is related to the vertical profile comparison against the radio soundings. Namely, the radio soundings from the Libuš meteorological station are assimilated into ERA5 data used for driving the WRF model, thus influencing the comparison and introducing the bias into evaluating the correctness of the ensemble members. On the other hand, after many statistical analyses were performed, no member significantly outperforms the rest of the ensemble with respect to correctness or performance in relation to the radio-sounding data. Moreover, the majority of the mesoscale models which could be potentially used for the preparation of the IBC for PALM have radio soundings assimilated directly, or indirectly as the WRF model through the ERA5 or other types of driving data making it a general problem for these types of studies.

Considering all listed limitations, we recognize this study to be reliable with plausible results. The plausibility of the results is confirmed by the vertical comparison against the radio-sounding observations.



# 6   Conclusions

The objective of this study was to address the following topics: i) constructing a "perfect" set of IBC for the PALM model, ii) sensitivity of the PALM model to the given IBC set, iii) performance of the PALM model based on the given IBC set.

i) The process of construction of a "perfect" set of IBC conditions from the WRF model for the purpose of driving the PALM model proved to be challenging. The evaluation of WRF outputs against observations has confirmed that the performance of any particular setting (parameterizations etc.) differs among variables; often there is a trade-off between performance in one variable against another one, e.g. temperature and wind speed. Also, the performance may change with the season.

ii) The differences between PALM simulations driven by different IBC's decrease and increase periodically throughout the simulation time, and the time patterns are different for different seasons. This behavior is, to some extent, consistent between different pairs of the PALM model outputs (driven by different IBCs) and it depends on the period of the day during the simulation time.

iii) As a general rule, PALM simulation conforms to the given set of IBC, and shows substantial consistency with them. Thus the largest part of errors may indeed originate in the mesoscale model. PALM model's performance, however, is not deteriorated when the given IBC set is farther from the real state of the atmosphere (i.e. observations) and it is not lagging when the IBCs are close to the observations. As seen from this experiment, there is a lot of place for bringing erroneous information into PALM through the initial and boundary conditions.

In order to fully assess the influence of the boundary conditions and PALM's sensitivity to them there is a need for long-term simulations followed by statistical evaluation, for different periods throughout the year. While being aware of the departures from reality introduced by the IBC, we may claim that PALM tends to attenuate the influence of possibly misspecified boundary conditions and its response to differences in the boundary conditions is fairly robust. Also, PALM has the capacity to reflect better the local processes (e.g., surface interactions and generation of turbulence) which is clearly an asset in the field of high-resolution modeling of the urban areas. These facts support a better confidence in results of PALM simulations performed with the aim of comparison of scenarios of urban development or mitigation strategies.

*Code and data availability.* The utilized source code, and description for PALM installation and usage guide, configuration files for the PALM model, input data for the PALM model, configuration files for WRF model, radio soundings used for comparison, as well as the scripts for pre- and post-processing are stored at Radović et al. (2023)

.





**Appendix A: WRF ensemble statistical analysis of the potential temperature and wind speed vertical profiles up to the height of 3000 m a.s.l.: RMSE-root mean square error; r-correlation coefficient.**

**Table A1.** February episode WRF ensemble statistical analysis of the potential temperature and wind speed vertical profiles up to the height of 3000 m a.s.l.: RMSE-root mean square error; r-correlation coefficient.

| WRF member ID | RMSE | | r | |
|---|---|---|---|---|
| | Potential temperature [K] | Wind speed [m s$^{-1}$] | Potential temperature | Wind speed |
| 01 | 0.9421 | 1.5101 | 0.9865 | 0.8668 |
| 02 | 1.1481 | 1.7281 | 0.9821 | 0.8287 |
| 03 | 0.9264 | 1.4956 | 0.9869 | 0.8686 |
| 04 | 0.9277 | 1.5028 | 0.9868 | 0.8683 |
| 05 | 0.9469 | 1.4454 | 0.9861 | 0.8770 |
| 06 | 1.1392 | 1.6705 | 0.9817 | 0.8385 |
| 07 | 0.9725 | 1.4121 | 0.9852 | 0.8815 |
| 08 | 0.9598 | 1.4028 | 0.9856 | 0.8833 |
| 09 | 0.9605 | 1.5063 | 0.9857 | 0.8677 |
| 10 | 1.1433 | 1.7416 | 0.9812 | 0.8266 |
| 11 | 0.9549 | 1.4917 | 0.9859 | 0.8696 |
| 12 | 0.9655 | 1.5049 | 0.9855 | 0.8681 |
| 13 | 0.9740 | 1.4426 | 0.9851 | 0.8774 |
| 14 | 1.1398 | 1.6738 | 0.9807 | 0.8380 |
| 15 | 1.0139 | 1.4087 | 0.9838 | 0.8821 |
| 16 | 1.0014 | 1.3978 | 0.9842 | 0.8843 |





**Table A2.** April episode WRF ensemble statistical analysis of the potential temperature and wind speed vertical profiles up to the height of 3000 m a.s.l.: RMSE-root mean square error; r-correlation coefficient.

| WRF member ID | RMSE | | r | |
|---|---|---|---|---|
| | Potential temperature [K] | Wind speed [m s$^{-1}$] | Potential temperature | Wind speed |
| 01 | 0.7491 | 1.5977 | 0.9803 | 0.8600 |
| 02 | 0.8327 | 1.7523 | 0.9756 | 0.8401 |
| 03 | 0.7704 | 1.7329 | 0.9791 | 0.8399 |
| 04 | 0.7716 | 1.7143 | 0.9790 | 0.8425 |
| 05 | 0.7429 | 1.6980 | 0.9796 | 0.8560 |
| 06 | 0.8360 | 1.8872 | 0.9745 | 0.8169 |
| 07 | 0.7991 | 1.7210 | 0.9762 | 0.8609 |
| 08 | 0.7837 | 1.7148 | 0.9771 | 0.8614 |
| 09 | 0.7631 | 1.5920 | 0.9796 | 0.8612 |
| 10 | 0.8442 | 1.7602 | 0.9748 | 0.8381 |
| 11 | 0.7824 | 1.7225 | 0.9785 | 0.8421 |
| 12 | 0.7965 | 1.7217 | 0.9773 | 0.8451 |
| 13 | 0.7619 | 1.7084 | 0.9786 | 0.8547 |
| 14 | 0.8534 | 1.8970 | 0.9735 | 0.8154 |
| 15 | 0.7991 | 1.7216 | 0.9763 | 0.8599 |
| 16 | 0.7891 | 1.7146 | 0.9768 | 0.8605 |





**Table A3.** July episode WRF ensemble statistical analysis of the potential temperature and wind speed vertical profiles up to the height of 3000 m a.s.l.: RMSE-root mean square error; r-correlation coefficient.

| WRF member ID | RMSE | | r | |
|---|---|---|---|---|
| | Potential temperature [K] | Wind speed [m s$^{-1}$] | Potential temperature | Wind speed |
| 01 | 0.7787 | 1.9274 | 0.9781 | 0.7863 |
| 02 | 0.8811 | 1.9461 | 0.9732 | 0.7882 |
| 03 | 0.8464 | 1.9318 | 0.9727 | 0.7850 |
| 04 | 0.9276 | 2.0069 | 0.9658 | 0.75787 |
| 05 | 0.8180 | 1.8098 | 0.9746 | 0.8132 |
| 06 | 0.9002 | 1.8483 | 0.9794 | 0.8069 |
| 07 | 0.8285 | 1.7491 | 0.9753 | 0.8266 |
| 08 | 0.8705 | 1.8002 | 0.9709 | 0.8164 |
| 09 | 0.7932 | 1.9413 | 0.9767 | 0.7822 |
| 10 | 0.8910 | 1.9600 | 0.9718 | 0.7824 |
| 11 | 0.8588 | 1.9408 | 0.9714 | 0.7815 |
| 12 | 0.9462 | 2.0057 | 0.9639 | 0.7579 |
| 13 | 0.8232 | 1.8222 | 0.9738 | 0.8095 |
| 14 | 0.9038 | 1.8667 | 0.9694 | 0.8005 |
| 15 | 0.8324 | 1.7741 | 0.9744 | 0.8193 |
| 16 | 0.8796 | 1.8117 | 0.9698 | 0.8121 |





**Table A4.** October episode WRF ensemble statistical analysis of the potential temperature and wind speed vertical profiles up to the height of 3000 m a.s.l.: RMSE-root mean square error; r-correlation coefficient.

| WRF member ID | RMSE | | r | |
|---|---|---|---|---|
| | Potential temperature [K] | Wind speed [m s⁻¹] | Potential temperature | Wind speed |
| 01 | 1.9236 | 3.5509 | 0.9520 | 0.6357 |
| 02 | 1.9918 | 3.4270 | 0.9472 | 0.6397 |
| 03 | 1.9649 | 3.5446 | 0.9505 | 0.6341 |
| 04 | 1.9805 | 3.4998 | 0.9490 | 0.6384 |
| 05 | 1.9025 | 2.6359 | 0.9539 | 0.6800 |
| 06 | 2.0551 | 2.6651 | 0.9442 | 0.6833 |
| 07 | 1.9465 | 2.5968 | 0.9517 | 0.6879 |
| 08 | 1.9288 | 2.6225 | 0.9521 | 0.6792 |
| 09 | 1.9414 | 3.5495 | 0.9509 | 0.6359 |
| 10 | 2.0195 | 3.4415 | 0.9456 | 0.6365 |
| 11 | 1.9779 | 3.5411 | 0.9496 | 0.6336 |
| 12 | 1.9769 | 3.3937 | 0.9479 | 0.6660 |
| 13 | 1.9217 | 2.6465 | 0.9527 | 0.6798 |
| 14 | 2.0790 | 2.6863 | 0.9427 | 0.6795 |
| 15 | 1.9608 | 2.6089 | 0.9506 | 0.6858 |
| 16 | 1.9467 | 2.6314 | 0.9508 | 0.6776 |





**Appendix B: PALM ensemble statistical analysis of the potential temperature and wind speed vertical profiles up to**
495 **the height of 3000 m a.s.l.: RMSE-root mean square error; r-correlation coefficient.**

**Table B1.** February episode PALM ensemble statistical analysis of the potential temperature and wind speed vertical profiles up to the height of 3000 m a.s.l.: RMSE-root mean square error; r-correlation coefficient.

| PALM member ID | RMSE | | r | |
|---|---|---|---|---|
| | Potential temperature [K] | Wind speed [m s$^{-1}$] | Potential temperature | Wind speed |
| 02 | 1.2507 | 1.8075 | 0.9806 | 0.7419 |
| 03 | 0.9038 | 1.5673 | 0.9875 | 0.7967 |
| 10 | 1.2437 | 1.8216 | 0.9791 | 0.7282 |
| 16 | 1.0178 | 1.4682 | 0.9836 | 0.8202 |

**Table B2.** April episode PALM ensemble statistical analysis of the potential temperature and wind speed vertical profiles up to the height of 3000 m a.s.l.: RMSE-root mean square error; r-correlation coefficient.

| PALM member ID | RMSE | | r | |
|---|---|---|---|---|
| | Potential temperature [K] | Wind speed [m s$^{-1}$] | Potential temperature | Wind speed |
| 05 | 0.6743 | 1.6602 | 0.9815 | 0.8485 |
| 09 | 0.6961 | 1.5758 | 0.9810 | 0.8504 |
| 14 | 0.7869 | 1.8698 | 0.9752 | 0.8053 |

**Table B3.** July episode PALM ensemble statistical analysis of the potential temperature and wind speed vertical profiles up to the height of 3000 m a.s.l.: RMSE-root mean square error; r-correlation coefficient.

| PALM member ID | RMSE | | r | |
|---|---|---|---|---|
| | Potential temperature [K] | Wind speed [m s$^{-1}$] | Potential temperature | Wind speed |
| 01 | 0.7419 | 1.9506 | 0.9779 | 0.7932 |
| 07 | 0.7398 | 1.7897 | 0.9784 | 0.8294 |
| 12 | 0.8426 | 2.0521 | 0.9705 | 0.7697 |





**Table B4.** October episode PALM ensemble statistical analysis of the potential temperature and wind speed vertical profiles up to the height of 3000 m a.s.l.: RMSE-root mean square error; r-correlation coefficient

| PALM member ID | RMSE | | r | |
| --- | --- | --- | --- | --- |
| | Potential temperature [K] | Wind speed [m s$^{-1}$] | Potential temperature | Wind speed |
| 01 | 1.8765 | 3.6162 | 0.9579 | 0.6469 |
| 05 | 1.8744 | 2.6507 | 0.9593 | 0.6898 |
| 07 | 1.9224 | 2.6195 | 0.9572 | 0.6957 |
| 14 | 1.9813 | 2.7447 | 0.9523 | 0.6777 |

**Appendix C: Minimum (min), average (avg), and maximum (max) three-day averaged differences for air temperature and wind speed for the selected WRF and PALM model outputs.**

**Table C1.** February episode minimum (min), average (avg), and maximum (max) three-day averaged differences for air temperature for the selected WRF and PALM model outputs.

| Air temperature [K] | WRF 16-10 | | | PALM 16-10 | | |
| --- | --- | --- | --- | --- | --- | --- |
| | min | avg | max | min | avg | max |
| Morning | 0.59 | 2.34 | 3.48 | -4.38 | 0.20 | 3.82 |
| Noon | 1.66 | 2.74 | 3.36 | -3.85 | 1.20 | 3.14 |
| Daytime | 1.50 | 2.38 | 2.87 | -2.27 | 1.01 | 2.52 |
| Nighttime | 0.31 | 2.02 | 3.11 | -4.66 | 0.04 | 2.98 |

**Table C2.** February episode minimum (min), average (avg), and maximum (max) three-day averaged differences for wind speed for the selected WRF and PALM model outputs.

| Wind speed [m s$^{-1}$] | WRF 03-02 | | | PALM 03-02 | | | WRF 16-10 | | | PALM 16-10 | | |
| --- | --- | --- | --- | --- | --- | --- | --- | --- | --- | --- | --- | --- |
| | min | avg | max | min | avg | max | min | avg | max | min | avg | max |
| Morning | -2.89 | -1.86 | -0.94 | -2.22 | -0.16 | 1.15 | -4.02 | -2.86 | -0.93 | -2.48 | -0.20 | 1.24 |
| Noon | -0.46 | -0.05 | 0.60 | -1.00 | 0.11 | 0.95 | -2.29 | -1.32 | 0.18 | -1.18 | -0.02 | 0.86 |
| Daytime | -1.17 | -0.49 | 0.49 | -0.92 | -0.01 | 0.53 | -2.82 | -1.74 | 0.06 | -1.49 | -0.10 | 0.68 |
| Nighttime | -3.23 | -1.90 | -0.95 | -1.98 | -0.15 | 0.75 | -4.27 | -2.86 | -1.11 | -2.32 | -0.26 | 0.57 |





**Table C3.** April episode minimum (min), average (Avg), and maximum (max) three-day averaged differences for air temperature for the selected WRF and PALM model outputs.

| Air temperature [K] | WRF 05-14 | | | PALM 05-14 | | | WRF 09-14 | | | PALM 09-14 | | |
|---|---|---|---|---|---|---|---|---|---|---|---|---|
| | min | avg | max | min | avg | max | min | avg | max | min | avg | max |
| Morning | -0.69 | -0.21 | 0.27 | -1.65 | 0.24 | 2.63 | -0.22 | -0.08 | 0.09 | -2.27 | -0.04 | 1.95 |
| Noon | 0.34 | 0.44 | 0.52 | -0.78 | 0.20 | 6.81 | 0.14 | 0.21 | 0.29 | -1.20 | 0.25 | 1.38 |
| Daytime | 0.18 | 0.22 | 0.28 | -0.90 | 0.14 | 3.68 | -0.01 | 0.03 | 0.12 | -0.90 | 0.03 | 0.61 |
| Nighttime | -0.45 | -0.28 | -0.13 | -2.33 | 0.02 | 2.30 | -0.68 | -0.26 | -0.05 | -2.41 | -0.17 | 2.44 |

**Table C4.** April episode minimum (min), average (avg), and maximum (max) three-day averaged differences for wind speed for the selected WRF and PALM model outputs.

| Wind speed [m s$^{-1}$] | WRF 05-14 | | | PALM 05-14 | | | WRF 09-14 | | | PALM 09-14 | | |
|---|---|---|---|---|---|---|---|---|---|---|---|---|
| | min | avg | max | min | avg | max | min | avg | max | min | avg | max |
| Morning | -0.17 | 0.57 | 1.28 | -1.78 | 0.02 | 1.22 | -1.28 | -0.69 | -0.41 | -0.65 | 0.10 | 1.24 |
| Noon | 0.08 | 0.46 | 0.67 | -0.87 | 0.03 | 0.71 | -0.30 | -0.07 | 0.10 | -0.54 | 0.27 | 1.08 |
| Daytime | 0.41 | 0.70 | 0.92 | -0.69 | -0.10 | 0.25 | 0.13 | 0.29 | 0.45 | -0.49 | 0.04 | 0.34 |
| Nighttime | 0.10 | 0.30 | 0.61 | -1.19 | -0.05 | 0.61 | 0.19 | 0.82 | 1.46 | -1.18 | 0.00 | 0.70 |

**Table C5.** July episode minimum (min), average (avg), and maximum (max) three-day averaged differences for air temperature for the selected WRF and PALM model outputs.

| Air temperature [K] | WRF 07-12 | | | PALM 07-12 | | |
|---|---|---|---|---|---|---|
| | min | avg | max | min | avg | max |
| Morning | -2.18 | -1.63 | -1.07 | -3.59 | -0.75 | 2.33 |
| Noon | -0.43 | -0.26 | -0.05 | -3.44 | -0.38 | 2.70 |
| Daytime | -0.71 | -0.45 | -0.20 | -2.32 | -0.54 | 1.11 |
| Nighttime | -4.39 | -2.60 | -0.76 | -4.49 | -1.43 | 1.43 |





**Table C6.** July episode minimum (min), average (avg), and maximum (max) three-day averaged differences for wind speed for the selected WRF and PALM model outputs.

| Wind speed [m s⁻¹] | WRF 01-12 | | | PALM 01-12 | | | WRF 07-12 | | | PALM 07-12 | | |
|---|---|---|---|---|---|---|---|---|---|---|---|---|
| | min | avg | max | min | avg | max | min | avg | max | min | avg | max |
| Morning | -0.02 | 0.48 | 1.01 | -0.93 | -0.06 | 0.88 | -0.89 | -0.31 | 0.53 | -1.14 | -0.05 | 0.71 |
| Noon | -2.36 | -1.97 | -1.10 | -1.11 | -0.04 | 0.92 | -1.98 | -1.70 | -1.01 | -1.05 | -0.03 | 0.91 |
| Daytime | -1.55 | -1.22 | -0.35 | -0.43 | 0.04 | 1.22 | -1.20 | -0.87 | -0.06 | -0.67 | -0.07 | 0.55 |
| Nighttime | -2.05 | -1.09 | -0.05 | -1.50 | -0.25 | 1.45 | -1.64 | -0.90 | 0.23 | -1.72 | -0.42 | 0.90 |

**Table C7.** July episode minimum (min), average (avg), and maximum (max) three-day averaged differences for MRT for the selected PALM model outputs.

| MRT [K] | PALM 01-12 | | | PALM 07-12 | | |
|---|---|---|---|---|---|---|
| | min | avg | max | min | avg | max |
| Morning | -1.32 | -0.55 | 0.10 | -1.64 | -0.71 | 0.11 |
| Noon | -1.75 | -0.50 | 1.59 | -1.72 | -0.22 | 1.50 |
| Daytime | -1.31 | -0.48 | 0.33 | -1.13 | -0.41 | 0.56 |
| Nighttime | -2.20 | -0.99 | 0.07 | -2.72 | -1.27 | -0.27 |

**Table C8.** July episode minimum (min), average (avg), and maximum (max) three-day averaged differences for PET for the selected PALM model outputs.

| PET [K] | PALM 01-12 | | | PALM 07-12 | | |
|---|---|---|---|---|---|---|
| | min | avg | max | min | avg | max |
| Morning | -2.34 | -0.54 | 0.85 | -2.72 | -0.74 | 1.06 |
| Noon | -6.22 | -0.61 | 6.43 | -5.41 | -0.37 | 6.14 |
| Daytime | -3.92 | -0.71 | 1.82 | -2.68 | -0.62 | 2.41 |
| Nighttime | -4.67 | -0.90 | -0.09 | -4.36 | -1.13 | 0.42 |





**Table C9.** October episode minimum (min), average (avg), and maximum (max) three-day averaged differences for air temperature for the selected WRF and PALM model outputs.

| Air temperature [K] | WRF 05-14 | | | PALM 05-14 | | | WRF 07-01 | | | PALM 07-01 | | |
|---|---|---|---|---|---|---|---|---|---|---|---|---|
| | min | avg | max | min | avg | max | min | avg | max | min | avg | max |
| Morning | -0.22 | -0.03 | 0.16 | -3.09 | -0.70 | 0.82 | 0.88 | 2.07 | 2.80 | -2.27 | 0.26 | 2.09 |
| Noon | -0.83 | -0.55 | -0.27 | -1.39 | -0.31 | 1.66 | 1.63 | 2.91 | 3.68 | -1.08 | 1.28 | 3.08 |
| Daytime | -0.54 | -0.34 | -0.10 | -0.87 | -0.19 | 0.67 | 1.15 | 2.14 | 2.80 | -0.34 | 0.89 | 2.01 |
| Nighttime | -0.95 | -0.81 | -0.67 | -1.76 | -0.39 | 0.82 | 0.63 | 1.54 | 2.22 | -1.64 | 0.37 | 1.42 |

**Table C10.** October episode minimum (min), average (avg), and maximum (max) three-day averaged differences for wind speed for the selected WRF and PALM model outputs.

| Wind speed [m s$^{-1}$] | WRF 05-14 | | | PALM 05-14 | | | WRF 07-01 | | | PALM 07-01 | | |
|---|---|---|---|---|---|---|---|---|---|---|---|---|
| | min | avg | max | min | avg | max | min | avg | max | min | avg | max |
| Morning | -0.21 | 0.43 | 0.70 | -2.12 | -0.16 | 0.91 | -1.11 | -0.44 | 0.44 | -1.54 | -0.03 | 1.44 |
| Noon | -0.40 | -0.12 | 0.01 | -0.80 | -0.17 | 0.56 | -0.19 | 0.01 | 0.27 | -0.80 | -0.11 | 0.38 |
| Daytime | -0.59 | -0.49 | -0.35 | -0.54 | -0.12 | 0.19 | -0.02 | 0.22 | 0.43 | -0.44 | 0.02 | 0.32 |
| Nighttime | -0.82 | -0.62 | -0.35 | -1.16 | -0.15 | 0.25 | -1.20 | -0.64 | -0.24 | -0.87 | -0.03 | 0.45 |





## Appendix D: Elevation map of the simulated domain.

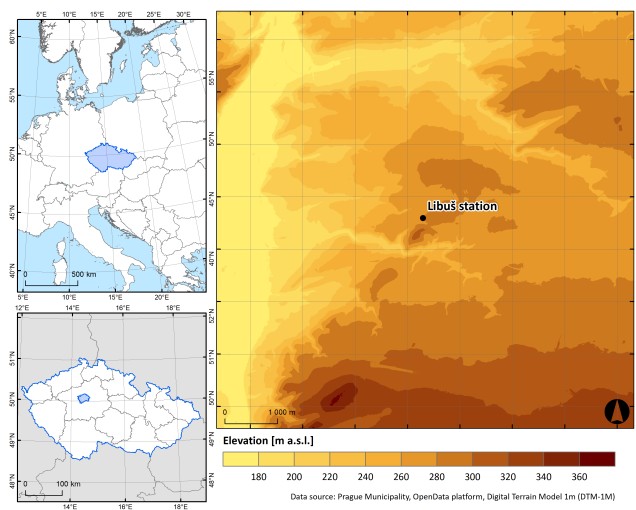

**Figure D1.** The location of the modeled domain in Europe (top left), and in the Czech Republic (bottom left) and the elevation map of the domain within the city of Prague with a Libuš station (WMO ID 11520) location (right).

## Appendix E: Experiment workflow diagram

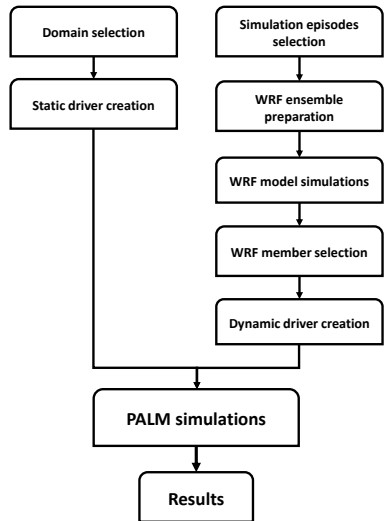

**Figure E1.** Experiment workflow diagram.



*Author contributions.* JRa, MBe, JRe, KE, PK, and VF designed the experiment. JRa performed the PALM model simulations, and KE performed the WRF model simulations. JRa, MBu, JG, and KE were involved in PALM and WRF data processing. JG and MBu were involved in geodata preprocessing. HŘ revised the text and was involved in topic discussions. JRa, MBe, and KE wrote the majority of the text, and all the co-authors contributed to discussions, text and revised the paper.

*Competing interests.* The authors declare that there is no competing interest present.

*Acknowledgements.* The PALM simulations were performed on the HPC infrastructure of the IT4I supercomputing center supported by the Ministry of Education, Youth and Sports of the Czech Republic through the e-INFRA CZ (ID:90254). The WRF simulations, as well as pre- and post-processing, were conducted on the HPC infrastructure of the Institute of Computer Science (ICS) of the Czech Academy of Sciences supported by the long-term strategic development financing of the ICS (RVO 67985807). This study is financially supported and conducted within the Turbulent-resolving urban modeling of air quality and thermal comfort - TURBAN project supported by the Norway Grants and Technology Agency of Czech Republic (grant no. TO1000219).





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
