# Peer review of "Challenges of constructing and selecting the "perfect" boundary conditions for the LES model PALM"

_Geoscientific Model Development, 2023_

## Author Comment (AC2)

**Reviewer 1**

This paper addresses the role of the driving model in an urban-area LES model that covers an 8 km square. In this study multiple configurations of the driving model with different biases were used to provide initial and boundary conditions to PALM. the main finding was that the biases do propagate into the PALM results, but PALM can also alleviate some of them. The main verification was wind and temperature for soundings and the surface, but the results were also compared to the spread of the driving model. Many results were presented in tables and supplementary plots.

Overall the paper is worth publishing because this type of downscaling is becoming a common tool for urban modeling that addresses some important problems affected by local details in built environments. It is useful to emphasize to what extent the driving model matters, and this is quantified by the methods adopted here. I will only have some minor comments to address, but I think some may be useful additional thoughts for consideration.

**Minor Comments**

**1. Table 1. It would have been useful to put the month with the episode number here.**

*Answer: Thank you for the suggestion, a new row has been added to the table with the dates of the episodes. Please see Table 1.*

**2. Line 135. Does the urban model include anthropogenic heating and a building energy model?**

*Answer: The simulation configuration employs the Building Surface Module (BSM; Resler et al., 2017, Maronga et al., 2020) with prescribed inner building temperature. The inner building temperature is prescribed according to the season, and heat exchange through walls/roofs is calculated accordingly. The exchange is modeled as a heat transfer between the inner building environment with a realistic inner temperature and the outer environment based on properties of walls and roofs (insulation, thermal capacity, thickness, etc.), and thus the heat energy produced inside the building (mainly heating or cooling) is considered. We use this method because we have more precise information about building construction than about installed heating/cooling systems and their operational conditions. We thus expect our approach to be less prone to large errors in our test cases than the Building indoor model (Pfafferott et al., 2021) since we do not have sufficient information for its complex calculation and due to that, we could bring significant uncertainties into the model energy balance.*

*1. Resler, J., Krč, P., Belda, M., Juruš, P., Benešová, N., Lopata, J., Vlček, O., Damašková, D., Eben, K., Derbek, P., Maronga, B., and Kanani-Sühring, F.: PALM-USM v1.0: A new urban surface model integrated into the PALM large-eddy simulation model, Geosci. Model Dev., 10, 3635–3659, https://doi.org/10.5194/gmd-10-3635-2017, 2017.*

2. Maronga, B., Banzhaf, S., Burmeister, C., Esch, T., Forkel, R., Fröhlich, D., Fuka, V., Gehrke, K. F., Geletič, J., Giersch, S., Gronemeier, T., Groß, G., Heldens, W., Hellsten, A., Hoffmann, F., Inagaki, A., Kadasch, E., Kanani-Sühring, F., Ketelsen, K., Khan, B. A., Knigge, C., Knoop, H., Krč, P., Kurppa, M., Maamari, H., Matzarakis, A., Mauder, M., Pallasch, M., Pavlik, D., Pfafferott, J., Resler, J., Rissmann, S., Russo, E., Salim, M., Schrempf, M., Schwenkel, J., Seckmeyer, G., Schubert, S., Sühring, M., von Tils, R., Vollmer, L., Ward, S., Witha, B., Wurps, H., Zeidler, J., and Raasch, S.: Overview of the PALM model system 6.0, Geosci. Model Dev., 13, 1335–1372, https://doi.org/10.5194/gmd-13-1335-2020, 2020.

3. Pfafferott, J., Rißmann, S., Sühring, M., Kanani-Sühring, F., and Maronga, B.: Building indoor model in PALM-4U: indoor climate, energy demand, and the interaction between buildings and the urban microclimate, Geosci. Model Dev., 14, 3511–3519, https://doi.org/10.5194/gmd-14-3511-2021, 2021.

**3. Line 225. Define MRT, PET and UTCI.**

*Answer: The following paragraph has been reformulated, with the definitions included, and additional information was added about the reasons for evaluating the thermal indices as suggested by Reviewer#2 in his comment on L225-276. For the new formulation please take a look at L263-L276.*

**4. Figure 2. Is it one point for each time and profile-average and each simulation, or each day? It is not clear how many points to expect to see here?**

*Answer: The additional information you've asked for in this comment we included in the manuscript. Please take a look at the paragraph from L300 to L307.*

**5. Supplementary Figures. It is noticed that the night time stable layer development is missed by both models, nor does PALM seem to show any improvement when in calm conditions it should be able to develop cooling if it was able. I am not sure that the driving model could be solely blamed for missing these and more likely both models are missing something. This should be discussed more even if it is just seen in Supplementary plots.**

*Answer: Thank you for pointing this out. We do agree with your opinion and have considered mentioning this issue in the manuscript. This is one of the first sets of experiments of such kind, and we would like to focus more on this issue in future research. This remains to be researched and it is yet to be seen if this behavior by PALM is systematic or is it not. In this instance, we can't "blame" either model based only on the data and simulations we have, thus, we would need more simulations. However, we've concluded that this effect is related to the PALM's dynamic core.*
*We included additional information about this comment in the manuscript. Please see L361-L367.*

**6. Line 318. "fringe-like pattern". Is this because the flow is from the west and the variation is along the west boundary?**

*Answer: We checked the data and concluded that the prevailing flow is from west to east (200-300 deg). We also checked the PALM original data (not differences which are presented in Figure 3b) and this pattern is present there as well. Our hypothesis for the formation of these waves is a consequence of the inflow boundary conditions, and for the details, we refer to the second reviewer's comment on Figure 3b.*

**7. Figure 3. We see a strong local signal which is presumably an entirely urban area, but it is hard to know because no maps show the WRF urban area. Could an outline be added for WRF urban grid cells?**

*Answer:*
*According to your suggestion, we've added WRF urban grid cells to Figures 3, and 4, and corresponding supplementary figures. For this purpose, we used the LU_INDEX WRF model variable which shows the landuse information. If the square is outlined with a black color the majority of the land use in that square is of urban type. If the square is not outlined with the black line, then the majority of the grid box area is not of urban type, albeit some of it can be.*

*Please see Figures 3, and 4 and supplementary figures with the description on them, and an additional description on L372-L373.*

**8. Figure 5b and others. It is interesting to note that the more urban cells have less difference due to driving data. I think this implies that the buildings are deterministic in some way in the local climate. This could be an interesting aspect of the results to discuss. What characteristics of urban density do these areas have?**

*Answer: In connection to your comment #11 we added a short comment in the manuscript on this feature. Please see the L470-L473.*

*In more detail, in Fig05b are differences between members 05 and 07, which means BouLac BEP urban physics (07) and with YSU BEM (12). It seems that both members have a similar parametrization for heterogeneous built-up areas (prevailing terrain is flat, there are only minor differences in building heights or density). The highest differences are located in forested areas of urban greenery (see classes 1410 and 3100 on Fig05a) in complex terrain, mostly on north-oriented valley sides. It means that both parameterizations have important differences in the parametrization of urban greenery and/or terrain orientation (probably the effect of different irradiation).*

**9. Figure 6. Some of these rows are redundant being just the reverse of other rows. It can be reduced to 6 rows without loss of information.**

*Answer: The unnecessary rows have been removed. In addition, as suggested by Reviewer#2 the font size has been increased, the displayed hours have been reduced to every three hours, and an additional x-axis has been added. Figure 7, and supplementary figures have been adjusted accordingly.*

**10. Line 405. As above, I think it is not completely clear that the local errors in both models can be blamed on the mesoscale model in cases where both miss the night-time stability. There is no indication that PALM has tried to correct this bias even in conditions of light wind where local effects should dominate.**

Answer: *This was more of a general comment, but we see how it might be confused with a discussion of our experiment. As per the suggestion of the second reviewer, the general text was moved to the introduction and only the discussion of the actual results was kept here. Please see L48-L56.*

**11. Line 412. Also related to this as stated earlier, dense urban areas appear less affected by the driving data, so in these areas the parameters may matter more.**

*Answer: Thank you for this observation, we added a comment about this in the manuscript (see also the answer to comment #8). Please see the L470-L473.*

**12. Line 444. Yes, WRF has recently added the local climate zone map which should improve urban morphology. LCZ could be mentioned here.**

*Answer: We have added information about the LCZ., please see L509-L510.*

**13. General comment. I think the large number of tables is not critical to the bulk of the paper while some soundings in the Supplementary data are. I think a better balance could be achieved. Or maybe some way could be found to condense the table information more.**

*Answer: We moved two tables summarizing the technical details from the manuscript text to the appendix. One is the table describing WRF model parameterizations, and the other one summarizes selected WRF ensemble members as the best/worst. They are now Appendix A and B respectively.*

---

## Author Comment (AC3)

**Reviewer 2**

**Overview**

The role of initial and boundary condition (IBC) uncertainty on high-resolution large-eddy simulations (LES) is a topic of high relevance for the LES modelling community, with many relevant open questions.

The manuscript investigates the effect of IBC uncertainty arising from the boundary layer physics parametrisations of the parent model on LES simulations with the state-of-the-art modelling system PALM.

In general, the model experiment the manuscript describes is systematic, is methodologically sound, uses relevant metrics and suits well in the scope of GMD. Studies like of that presented in the manuscript, investigating different aspects of IBC sensitivity in high-resolution LES, are much welcomed.

However, there are some issues stated below which should be addressed prior to publication. In addition, a careful review of the language and proofreading is required. Given that the issues are addressed by the authors, the manuscript can be recommended for publication in GMD.

**Major comments**

**Major comment #1:**

The manuscript addresses a certain subset of the open questions on IBC-related uncertainties. More specifically, it analyses those arising from incomplete description of boundary layer physics by the WRF model schemes. There are still a lot more open questions related to IBCs in LES though, such as sensitivity to errors in synoptic scale forcing, improper representation of highly dynamic features such as fronts by temporal interpolation of discrete output time steps of the parent model and sensitivity of the LES-resolved dynamics on the domain size, to mention some. The limitations of the study regarding remaining open questions should be discussed further and more clearly. Also in this context, rephrasing the aims and objectives presented in the introduction and in the conclusions to better reflect the research questions that can be addressed using the described model experiment would benefit the manuscript. Similarly, a more specific title for the manuscript, reflecting that only a certain subset of the open questions related to optimal IBCs for LES was studied, could be considered.

*Answer: We reformulated some parts of the introduction, and added a short specification of the subset of problems we are discussing (very brief not to repeat the Limitations section).*
*Please see L97-L107 for a better representation of our aims and L490-L494; L498-L499; and L500 for stating all the limitations this study has. Also, we moved the general comment previously on L400-408 from the discussion to the introduction which is on the lines: L48-L56.*
*Also with regard to your third major comment, the manuscript title has been changed, and "Initial" in "Initial and Boundary Conditions (IBC)" has been omitted in the title and through the text as well. The reason for this is that the influence of the initial conditions is not treated*

*separately in this experiment (eg. by having a fixed set of BCs and only varying ICs) and as you correctly point out, the influence of IC will decrease over time.*

**Major comment #2:**

Given that the specific focus of the manuscript is IBC uncertainty, a detailed description of how exactly the offline IBCs were processed from the WRF model outputs should be added to the manuscript. As the IBCs are the central theme for the manuscript, this should be included in the manuscript itself, especially because PALM's own documentation is unfortunately not that detailed on the technicalities. The description should include, but not limit to, description of e.g. the spatial and temporal interpolation methods, the temporal resolution of WRF output data, topography matching, applied mass-flux correction etc. Also, the limitations of the used synthetic turbulence generator should be discussed. This would allow for more in-depth critical discussion of the processing pipeline in the review and also in the case that the topic is discussed further in later studies. It would also make the process more clear to the readers, without need of browsing through prior literature and incomplete documentation.

*Answer: We agree that a better description of the IBC preparation method is suitable and we have added a more detailed description as suggested by the reviewer. Please see L190-L215.*

**Major comment #3:**

I don't see that LES domain size sensitivity is discussed in the manuscript. The domain size dictates how much freedom the LES-based dynamics and PALM model physics have to diverge from the IBCs, and is thus a crucial component for IBC sensitivity. Has a domain size sensitivity test been performed? I regard domain size sensitivity a higher concern than the sensitivity to grid resolution, as it has more direct influence on the analysed metrics, study objectives and research questions in this particular case. Furthermore, the influence of initial conditions on the LES results decreases over simulation time, and I don't see this being discussed in the manuscript.

*Answer: We did not perform any sensitivity tests on domain size for this experiment in order to keep the study as concise as possible, by fixing all other variables besides the IBC. The influence of domain size for nested simulations has been the subject of several studies in limited-area NWP. regional climate modeling and also LES and as you correctly point out, the larger the domain, the more freedom the nested model has to develop "its own" meteorology. We added this concern to the limitations of the study and plan to explore it more in further research. Please see L500 and a short description with references in our aims: L102-L107.*

*We provide here a brief overview of the papers. To the best of our knowledge, parameter studies for urban flow models are rare. Usually, they deal with simplified domains and assess RANS models (e.g., Ramponi, R. and Blocken, B., 2012 or Abu-Zidan et al., 2021, Crank et al., 2018 ). Studies like Lamaakel et al. (2023) or Ovchinnikov et al. (2022) investigate the domain size sensitivity of the LES simulations for specific situations such as precipitating shallow cumulus convection and convective boundary layer, respectively. In addition, Ramponi et al., 2012 conducted a sensitivity study using only the RANS method, and among other parameters*

*explored the sensitivity of the domain size for a generic isolated building. This study stresses that the largest impact on their simulation is information about the inlet turbulent kinetic energy profile and not the domain size.*

1. *Abu-Zidan, Y., Mendis, P., and Gunawardena, T.: Optimising the computational domain size in CFD simulations of tall buildings, Heliyon, 7, e06723, https://doi.org/10.1016/j.heliyon.2021.e06723, 2021.*
2. *Ai, Z. T. and Mak, C. M.: Modeling of coupled urban wind flow and indoor air flow on a high-density near-wall mesh: Sensitivity analyses and case study for singlesided ventilation. Environ. Modell. Softw., 60, 57–68, https://doi.org/10.1016/j.envsoft.2014.06.010, 2014.*
3. *Crank, P. J., Sailor, D. J., Ban-Weiss, G., and Taleghani, M.: Evaluating the ENVI-met microscale model for suitability in analysis of targeted urban heat mitigation strategies, Urban Clim., 26, 188–197, https://doi.org/10.1016/j.uclim.2018.09.002, 2018.*
4. *Lamaakel O., Venters R., Teixeira J., and Matheou G.: Computational Domain Size Effects on Large-Eddy Simulations of Precipitating Shallow Cumulus Convection, Atmosphere, 14, 1186, https://doi.org/10.3390/atmos14071186, 2023.*
5. *Ovchinnikov, M., Fast, J. D., Berg, L. K., Gustafson, W. I., Jr., Chen, J., Sakaguchi, K., and Xiao, H.: Effects of Horizontal Resolution, Domain Size, Boundary Conditions, and Surface Heterogeneity on Coarse LES of a Convective Boundary Layer. Monthly Weather Rev., 150, 1397-1415. https://doi.org/10.1175/MWR-D-21-0244.1, 2022.*
6. *Ramponi, R. and Blocken, B.: CFD simulation of crossventilation for a generic isolated building: Impact of computational parameters, Build. Environ., 53, 34–48, https://doi.org/10.1016/j.buildenv.2012.01.004, 2012.*

**Specific comments**

**L32-40: I don't think there exists "the" "large eddy simulation for urban environments". The current phrasing lifts uDALES above other urban-scale LES models, incl. PALM, for an unknown reason. I think the authors could consider how the whole paragraph could be revised for better communication of the relevant information.**

*Answer: The following paragraph has been reformulated, please see L31-34.*

**L56: The concept of "PALM-4U" is not introduced prior to this. I recommend using just "PALM" or "PALM model system" throughout the manuscript, as PALM-4U components are part of PALM. In any case, the terminology should be consistent and clear to the readers.**

*Answer: Thank you for pointing this out. We removed PALM-4U and we reset the naming of the model. The model is introduced as the PALM model system in the abstract (see L2), and from that point on only PALM is used. PALM-4U is mentioned in section 2.2.1 where the model configuration is described, please see L146.*

**Table 1: The way to report the wind speeds here is a bit ambiguous. Consider reporting the mean geostrophic wind speed for 3000 m (from the pressure gradient, assuming it doesn't change a lot) and e.g. an interpercentile range of 25-75 for the 10 m wind speed.**

*Answer: Thank you for your suggestion. Since we have the radio-sounding data available we've calculated interpercentile range and the means for the 10 m and 3000 m wind speeds and put the information in Table 1 accordingly. Please see Table 1 for the changes.*

**L126-137: It should be clearly separated which of the facts are specific to the model itself and which are specific to the used model configuration. E.g. PALM implements three different models for subgrid-scale turbulent diffusion, and one is selected for use by the authors.**

*Answer: We have modified the paragraph. Please see L141 and L152.*

**L135: The PALM documentation refers to the "building surface model" as an "urban surface model", abbreviated USM, not BSM. Is there a reason why the authors opted to use a different name and abbreviation than the official documentation? Admittedly, the authors' choice actually describes the model better, but this might confuse those familiar with PALM.**

*Answer: This model naming has a historical background; in 2017, a large extension called PALM-USM was introduced (Resler et al., 2017). It included the radiative transfer model, the model for urban surface energy balance, and changes to the plant canopy model. The original concept included selected urban-related micro-meteorological processes, so USM was an appropriate abbreviation. Later, in 2020, the module containing the PALM-USM extensions was split into several modules, which were expanded and improved significantly (overview in Maronga et al., 2020). The surface energy balance was put into the module called USM, which was later officially renamed to BSM as the urban ground surfaces like pavements were moved to the land surface model (LSM), but the original name USM is still used in several places, e.g. in the name of the source file.*

*1. Resler, J., Krˇc, P., Belda, M., Juruš, P., Benešová, N., Lopata, J., Vlˇcek, O., Damašková, D., Eben, K., Derbek, P., Maronga, B., and Kanani-Sühring, F.: PALM-USM v1.0: A new urban surface model integrated into the PALM large-eddy simulation model, Geosci. Model Dev., 10, 3635–3659, https://doi.org/10.5194/gmd-10-3635-2017, 2017.*

*2. Maronga, B., Banzhaf, S., Burmeister, C., Esch, T., Forkel, R., Fröhlich, D., Fuka, V., Gehrke, K. F., Geletiˇc, J., Giersch, S., Gronemeier, T., Groß, G., Heldens, W., Hellsten, A., Hoffmann, F., Inagaki, A., Kadasch, E., Kanani-Sühring, F., Ketelsen, K., Khan, B. A., Knigge, C., Knoop, H., Krč, P., Kurppa, M., Maamari, H., Matzarakis, A., Mauder, M., Pallasch, M., Pavlik, D., Pfafferott, J., Resler, J., Rissmann, S., Russo, E., Salim, M., Schrempf, M., Schwenkel, J., Seckmeyer, G., Schubert, S., Sühring, M., von Tils, R., Vollmer, L., Ward, S., Witha, B., Wurps, H., Zeidler, J., and Raasch, S.: Overview of the PALM model system 6.0, Geosci. Model Dev., 13, 1335–1372, https://doi.org/10.5194/gmd-13-1335-2020, 2020.*

**L140-142: It could be added how this spinup is forced. The clear-sky assumption can be a source of bias for the surface fluxes due to excess heat storage if the real-life conditions have been cloudy.**

*Answer: We modified the text to better express that we employ the built-in feature of the PALM model and to notice the possible consequences of the spin-up simplifications. Please see L153-L160.*

*We provide here a piece of additional information. The spin-up feature in the PALM is designed as a cheap and simple way to create the first approximation of the realistic spatial distribution of the thermal conditions in the individual layers of heat storage (grounds, walls, and roofs). The process is simplified radically without the utilization of any external time-dependent meteorological information. Only the energy balance-related processes are employed, near-surface flow is calculated from the initial wind profile, and air temperature is estimated by a simple sinusoidal diurnal cycle based on the cosine of the solar zenith angle that is calculated based on the geographical location, time, and two user prescribed parameters (maximal and minimal temperature), and also radiation is estimated by simple clear-sky radiation model (see Maronga et al., 2020). Also, the real length to establish proper thermal conditions in the deep layers is usually longer than the feasible length of the spin-up simulation. The reviewer is definitely right that these simplifications can lead to only approximate estimation of the storage layer temperatures and can be burdened by biases but it is still a way to enable a basic spatial adjustment of the storage temperatures which were initialized from the coarse mesoscale model. When our focus is on the near-surface processes, we need to perform additional spin-up simulation with full model dynamics and energy balance. In our case, the vertical profiles are the primary focus of our study and they are very little influenced by the detailed structure of the temperature on the ground as proved by our internal test (not shown in the manuscript). On the other hand, the implementation of an additional parameter characterizing the cloud covering in the model should be easy and could help to avoid larger biases in cloudy conditions. We thank the reviewer for this idea and we will try to propose such an improvement of the model.*

**Figure 1: I would suggest using LCZs instead of the Urban Atlas codes for better interpretability. Also, it would be beneficial to show the boundaries of the nested WRF domains and the PALM domain in the figure. Usage of LCZs complemented with the domain boundaries would assist the readers in interpreting the spatial patterns in the results.**

*Answer: Thank you for your recommendation. LCZs are applicable for a lowest resolution between 100-300m (it is climatological classification). LCZ classification was developed for a station classification primarily; see Stewart 2011 (https://circle.ubc.ca/handle/2429/38069) or Stewart & Oke, 2012 (https://doi.org/10.1175/BAMS-D-11-00019.1). If we are interested in the scale of tens of meters, the LCZ concept is 'too rough' for this analysis. We used the Urban Atlas 2018 geodatabase as an input for a detailed PALM simulation, so we used this land cover layer in Figure 1. Moreover, the LCZ concept is not easily applicable for heterogeneous city structures as we have in the Czech Republic, in Prague or Brno more specifically*

*(see Geletič et al., 2016, 2018 ([https://doi.org/10.3390/rs8100788](https://doi.org/10.3390/rs8100788) or [https://doi.org/10.1016/j.scitotenv.2017.12.076](https://doi.org/10.1016/j.scitotenv.2017.12.076)).*
*WRF model's outer and inner domain extent has been added to Figure 1. Please take a look at Figure 1.*

**L176: With STG, the generated turbulence is not only "started" artificially, like when imposing disturbances until a certain disturbance energy level is reached, but rather continuously imposed on the non-turbulent BC. The physical basis/principle of the Xie and Castro STG method could be described with a couple of phrases here as well.**

*Answer: The following paragraph has been reformulated. Please see L210-L214.*

**L181: The information on domain extents/sizes is missing.**

*Answer: Thank you for pointing this out, we've added information about the WRF domain sizes. Please see L218-L219.*

**L194: Abbreviation NOAH LSM should be defined and cited even though it is just an example here.**

*Answer: The Abbreviation is corrected and it is as follows: NOAH Land Surface Model (NOAH LSM). The citation for the NOAH LSM scheme has been added to the manuscript, please see L232-L233.*

**L225-227: I think it would be beneficial for the readers if the authors would define MRT, PET and UTCI. This would give the readers a better understanding of what factors are considered in the given metric (thus a better understanding of the potential error sources). It would be also good to reason why these particular metrics were selected for analysis and comparison.**

Answer: *The following paragraph has been reformulated, with the definitions included, and additional information was added about the reasons for evaluating the thermal indices. Please see L263-L276.*

**L246-247: Are instantaneous values here used for the models as well or did the authors perform any temporal averaging for these?**

*Answer: When it comes to the PALM model outputs, the data used for the vertical profiles are taken from the 10-minute averaged PALM files. Additional time averaging was not performed. The values for statistics and the vertical profile comparison are taken at the exact times to match the sounding times (e.i. 00:00, 06:00, 12:00). The WRF model vertical profiles (both for statistical evaluation and qualitative evaluation) are instantaneous values taken from the 1-h outputs.*

*According to Reviewer#1 comment on Figure 2, we've added additional information on this topic, so please take a look at the L300-L307.*

**FIgure 2: It would be interesting to see Taylor diagrams of the results. In general, I find Taylor diagrams highly useful in model error comparisons. Especially in this case, as a turbulence-resolving model PALM can be assumed to be more "active" than WRF.**

*Answer:  Thank you for the idea. Taylor plots were generated (see Supplement, Figure S37) and a short text was added, please see L315-L317.*

**Figure 3, Tables 4-5: A suggestion: the July episode could be added to Figure 3 as a second column. Also the mean values could be then included in the figure. Then Tables 4 and 5 could be then moved to the appendix**.

*Answer: Thank you for your suggestion. We have considered the changes. However, we prefer the current setup of figures because when combined as per your suggestion, the final figure contains too many subplots and information which becomes difficult to read.*

**Figure 3b: The wave-like pattern is interesting here. Was Rayleigh damping applied above the BL? If not, did the authors ensure that there are no (artificial) gravity waves in the simulations?**

Answer: *The RD was not applied in our simulations. We are interested here in the near-surface velocity (10 m wind speed) and the wave pattern is the strongest close to the inflow boundary considering the prevailing westerly wind direction. There could be the strongest influence on the flow from the inflow boundary conditions. The waves that can be generated at the inflow in stable stratification were demonstrated, e.g., by Bodnár et al. (2021). The RD would not be able to prevent the formation of such waves.  The referenced paper also discusses the influence of the top boundary condition.*
*In addition, in our previous tests, the RD did not help with this problem, there was some influence on the results but it was weak and it did not change the character of the simulations.*

 *Tomáš Bodnár, Philippe Fraunié, Petr Knobloch, Hynek Řezníček. Numerical evaluation of artificial boundary condition for wall-bounded stably stratified flows. Discrete and Continuous Dynamical Systems - S, 2021, 14(3): 785-801. doi: 10.3934/dcdss.2020333*

**Figure 5: I would use LCZs here as well instead of the Urban Atlas codes, as they are more well known in the urban climate community and include fewer categories.**

*Answer: Thank you for your suggestion. We preferred to use Urban Atlas 2018 data to represent the land cover layer in Figure 5 for the same reasons described in our response to your comment on Figure 1.*

**Figure 6: This is a bit hard to read. Please consider increasing the font size so that it would be closer to that of the other figures. The spacing between the x-axis ticks could be increased, e.g. every 3 or 6 hours. The upper panel could also have its own x-axis, as now at least I had to rely on the aid of a ruler.**

*Answer: The font size has been increased, an additional x-axis has been added, the spacing between hours has been reduced, and as suggested by Reviewer#1 the number of rows has been reduced to 6. Figure 7, and supplementary figures have been adjusted accordingly.*

**L400-408: I don't think this discussion is tied well enough to the results. This could be moved either to the introduction with appropriate references or it should be better tied to the results.**

*Answer: Thank you for your comment, we do agree with it; this is a general comment and its inclusion here might be confusing, so it was moved to the introduction. Please see L48-L56.*

**L410-412: I would argue that the magnitude of sensitivity really depends on the metric you study. The diagnostic surface outputs are naturally very sensitive to surface parameters. However, these can't be alone used to assess the overall sensitivity of the PALM.**

*Answer: Thank you for your comment. We do agree with your opinion. In this case, we wanted to highlight the need for high-quality IBC data, and that introducing the error through the IBC could have a higher impact on the PALM outputs than introducing the error in the surface parameters would have in the case of model validation. Of course, in the optimal scenario, all of the input data should have as little as possible erroneous information.*

**L421-423: While in principle I do agree that the offline nesting performance across the mesoscale models could be studied and it is beneficial to have interfaces to as many models as needed by the users, I think it should be clearly mentioned here what kind of contribution to our pre-existing knowledge could be expected from this.**

*Answer: The necessity for coupling the PALM model to other mesoscale models stems from the fact that different mesoscale models have different (better or worse) performances for a certain period (summer, winter, etc. ). Thus, for a certain situation, the WRF model outputs could be optimal drivers for a given PALM simulation, while for some others, for example, ICON can be the one to use. In addition, different mesoscale models have different parameterization schemes and physics representations which influence how well the model represents the atmosphere over a certain area or a certain weather situation. All things considered, before performing a PALM simulation, one can test the quality of different mesoscale model outputs and pick the one that agrees the best with the observations for a certain time period and use that model's outputs as a driver for PALM.*
*Please see line L480-L483.*

**Section 5: A bit of a repetition from the general comments, but the sensitivity of the results on (PALM) domain size is not discussed here. The larger the PALM domain, the more freedom the LES-based dynamics have to deviate from the IBCs.**
**Furthermore, as initial conditions are included as a theme in the manuscript, the sensitivity on how long the LES model is run before the data is obtained from it is not discussed. If it is the case that these have not been studied in this study as it seems, they are a significant limitation for the study and should therefore be included in this list.**

*Answer: We've modified the text and put the limitations of this study as suggested in major comments 1 and 3 for the domain size, we refer to the responses in major comments 1 and 3. For the initial conditions we've added the additional limitation, please see L498-L499.*

**L451-454: I would not recommend making statements that are not either well-known, shown in the manuscript, included in the supplementary data nor published prior.**
*Answer: We have removed the statements not shown in the manuscript and supplementary data. Please see L490-L491.*

**L455-460: It is great to see that a leakage of the validation data into the model inputs is recognized here, even though the consequent bias in this case can be safely assumed to be minimal.**

*Answer: Thank you for your comment.*

**Technical corrections**

**L2: Should be "the PALM model system" instead of "PALM". Also, PALM is not an abbreviation anymore, but an independent name.**

*Answer: We've modified all the inconsistencies about naming the PALM model. At the L2 we put PALM model system, and from that point on, we've used only PALM.*

**L58: Underscore (_) is unnecessary.**

*Answer: The underscore in the WRF_interface has been removed. Please see L65.*

**Table 1: "scattered" has an extra "s".**

*Answer: An extra "s" has been removed from the word "scattered" in Table 1. Please take a look at the Table1.*

**L633: The Zenodo link is incorrect and leads to the 100 m LCZ dataset.**

*Answer: The correct link that leads to the experiment's additional materials has been added to the manuscript. Please see L710.*